# Remote sensing and modeling analysis of the extreme dust storm hitting Middle East and Eastern Mediterranean in September 2015

**Solomos Stavros[1], Albert Ansmann[2], Rodanthi-Elisavet Mamouri[3], Ioannis Binietoglou[1,5], Platon Patlakas[4], Eleni Marinou[1,6] and Vassilis Amiridis[1]**

[1]Institute for Astronomy, Astrophysics, Space Applications and Remote Sensing (IAASARS), National Observatory of Athens, Athens, Greece
[2]Leibniz Institute for Tropospheric Research, Leipzig, Germany
[3]Cyprus University of Technology, Department of Civil Engineering and Geomatics, Limassol, Cyprus
[4]School of Physics, Division of Environment and Meteorology, University of Athens, Athens, Greece
[5]National Institute of R&D for Optoelectronics, Magurele, Ilfov, Romania
[6]Laboratory of Atmospheric Physics, Physics Department, Aristotle University of Thessaloniki, 54124, Thessaloniki, Greece

**Abstract** The extreme dust storm that affected Middle East and the Eastern Mediterranean in September 2015 resulted in record-breaking dust loads over Cyprus with aerosol optical depth exceeding 5.0 at 550 nm. We analyze this event using profiles from the European Aerosol Research Lidar Network (EARLINET) and the Cloud-Aerosol Lidar and Infrared Pathfinder Satellite Observation (CALIPSO) as well as geostationary observations from the Meteosat Second Generation - Spinning Enhanced Visible and Infrared Imager (MSG-SEVIRI) and high resolution simulations with the Regional Atmospheric Modeling System (RAMS). The analysis of modeling and remote sensing data reveals the main mechanisms that resulted in the generation and persistence of the dust cloud over Middle-East and Cyprus. A combination of meteorological and surface processes is found: (a) the development of a thermal low at the area of Syria that results in unstable atmospheric conditions and dust mobilization at this area; (b) the convective activity over Northern Iraq that triggers the formation of westward moving haboobs that merge with the previously elevated dust layer; and (c) the changes in land use due to war at the areas of Northern Iraq and Syria that enhances dust erodibility.

## 1. Introduction

A record dust storm affected the entire Middle East and Cyprus in September 2015. Remote sensing and in-situ measurements of Arabian dust from this episode during 7-11 September 2015 are presented by Mamouri et al. (2016) for the station of Limassol (34.7°N, 33°E). As reported in this article, the extreme amounts of dust over Middle East and the Eastern Mediterranean originate from the desert and arid areas of Syria and Northern Iraq. Triggered by this work, we analyze here the main processes that resulted in the mobilization of dust due to a combination of cyclonic flow and haboob formation.

Haboobs are local and mesoscale atmospheric density currents that mobilize huge amounts of dust and create a propagating dust wall extending up to 2-3 km in the troposphere (Knippertz et al., 2009; Solomos et al., 2012). These systems are well known by local populations in desert and arid areas worldwide due to their devastating impact on visibility and human health (e.g. Schepanski et al., 2009; Emmel et al., 2010; Dempsey 2014; Pantillon et al., 2016). Haboobs are formed by the evaporation (and melting) of hydrometeors as they fall through warm, unsaturated air below the cloud base of convective clouds. The energy required for these phase changes (latent heat) generates cooled downdrafts. When the downdrafts hit the surface they spread out due to their enhanced density compared with the ambient air. These convective outflow boundaries are turbulent and gusty and when they travel over bare soil and desert areas sediment can be lifted, creating a propagating dust wall. The scale of the processes that participate in the generation of such atmospheric density currents ranges from synoptic down to mesoscale and local. As a result, haboobs and their effects in weather and air-quality cannot be resolved by the coarse global model resolutions (Marsham et al., 2013). Moreover, haboobs are usually generated over remote arid areas where no in-situ networks are present and in-site dust-storm measurements can only be obtained during field campaign experiments (e.g. SAharan Mineral dUst experiment (SAMUM) 1 & 2, Ansmann et al., 2011; FENNEC, Ryder et al., 2015). Following these limitations, most of the efforts for the studying and forecasting of such intense dust episodes rely on passive and active remote sensing (e.g. Moderate Resolution Imaging Spectroradiometer (MODIS), European Aerosol Research Lidar Network (EARLINET), Cloud-Aerosol Lidar and Infrared Pathfinder Satellite Observation (CALIPSO)) and on high resolution modeling simulations. Assimilation of satellite derived Aerosol Optical Thickness (AOT), has been shown to improve the dust forecasts in global models especially

for the long range transport (Benedetti, et al., 2009); however this approach cannot be easily adopted for the description of haboobs. The reason is that the convective events and the associated wind gusts are not properly resolved at coarse model resolutions. As a result, assimilating the satellite AOT over an inaccurate meteorological field does not improve the dust forecast.

A variety of studies on haboobs have been performed worldwide. For example Knippertz et al. (2009); Reinfried et al. (2009); Solomos et al. (2012); Roberts and Knippertz (2014) analyzed the physical processes that lead in severe haboob formation in Sahara.  Bou Karam et al. (2008) showed the contribution of the east Atlantic monsoon flow and the associated mesoscale convective systems (MCS) in dust elevation along the Sahel. Vukovic et al. (2014) described the severe convective dust storm that hit Phoenix Arizona in July 2011. Asian haboobs from the Taklimakan and Gobi deserts are described and simulated in Takemi (1999, 2005). All these studies agree in the complexity of the various physical processes at multiple atmospheric scales that govern the generation and lifetime of these systems. Apart from their devastating effects at local and near surface scales, such events may also contribute to the free-troposphere dust burden in several ways: First, entrainment of dust particles in the free troposphere takes place at the turbulent region of the density current head (Takemi, 2005; Solomos et al., 2012); Second, they trigger secondary convective cells along their pathways that may evolve to synoptic scale dust events and third, dust residuals remain aloft after the cold pool declines.

The current article is the second part (Part 2) in a series of articles on the September 2015 extraordinary dust storm in Middle East and Eastern Mediterranean. In Part 1, Mamouri et al. (2016) presented a detailed analysis of remote sensing and in-situ monitoring of the event over Cyprus.  The formation of similar events is not fully understood and we use this unique episode to elucidate the mechanism of dust production in this understudied region. Therefore EARLINET measurements over Cyprus along with CALIPSO and MSG observations are used to fine tune the Regional Atmospheric Modeling System (RAMS) simulations and explain the physical processes that resulted in this haboob-driven dust storm. We focus our analysis on the first two days of the event (6 and 7 September 2015) when the extraordinary dust-storm was generated. To the best of our knowledge this is the first detailed modeling and remote sensing study to describe a Middle East haboob. The modeling and measurement techniques for the analysis are presented in Section 2. Section 3 includes the model results, the remote sensing and the investigation of the atmospheric processes that led in the formation of the dust episode. Conclusive remarks and discussion are

presented in Section 4.
**2. Instruments and models.**

**2.1 Remote sensing**
**2.1.1. EARLINET**
The lidar station at Limassol (34.7° N, 33° E; 23 m above sea level, a.s.l.) is part of the European
Aerosol Research Lidar Network (EARLINET: Pappalardo et al., 2014). The EARLINET lidar network is
widely used for aerosol characterization and particularly for dust characterization studies (Mona et
al., 2012). Details on the lidar station equipment and the retrieval algorithms are given in Mamouri
et al. (2016). Dust mass concentration profiles are obtained from the dust optical properties
following the methodology proposed by Ansmann et al. (2012).

**2.1.2. CALIPSO**
The Cloud-Aerosol Lidar with Orthogonal Polarization (CALIOP) is the principal instrument on board
the CALIPSO satellite. CALIOP is a standard dual-wavelength (532 and 1064 nm) backscatter lidar,
operating a polarization channel at 532 nm (Winker et al., 2009) and it has been acquiring high-
resolution profiles of the attenuated backscatter signal at 532 and 1064 nm along with polarized
backscatter signal in the visible channel since 2006. After calibration and range correction of the
lidar backscatter signals (Level 1 CALIPSO product), cloud and aerosol layers are identified and
aerosol backscatter and extinction coefficient profiles at 532 and 1064 nm are retrieved as part of
the Level 2 CALIPSO product. The CALIPSO algorithms are described in detail by Winker et al. (2009).
In this study, we utilize L2 version 3 Aerosol and Cloud profiles product at a horizontal resolution of
5 km and vertical resolution of 60 m (in altitudes up to 8 km above sea level). In extreme haboob
events, where the optical signal is very high, it is possible for the algorithm to wrongly attribute a
dust layer as a cloud. In order to address this issue and fully understand the observed scene we use
collocated information derived from MSG-SEVIRI (see sect. 2.1.3). In the two CALIPSO cases used
here, MSG-SEVIRI RGB images confirmed that CALIPSO overpasses was cloud free, hence we classify
both aerosol and cloud categorized CALIPSO observations as aerosol.
Moreover, both of the cases have significantly high particle depolarization ratio values, which is a
signature of pure dust scenes. In order convert the dust extinction coefficient from CALIPSO into
dust mass concentration, we follow the methodology proposed by Ansmann et al. (2012) using the
conversion parameter of desert dust that is proposed in Mamouri and Ansmann (2017). For this
case study we use a lidar ratio of 40 sr that is typical for Middle East dust (Mamouri et al., 2013).
The overall uncertainty in the estimated dust mass concentrations is 20-30%.

**2.1.3. MSG-SEVIRI**
The Meteosat dust RGB composite is produced from a combination of three infrared channels of
SEVIRI: IR12.0-IR10.8 (red), IR10.8-IR8.7 (green), and IR10.8 (blue). The channel combination and
visualization parameters (Table 1) were chosen to maximize the visual contrast between the hot
desert surface and lofted dust particles (Lensky and Rosenfeld, 2008). During daytime, the hot
desert sand, made up from large quartz particles, appears white/blue due to the large difference in
emissivity of IR10.8 and IR8.7 channels (green), high temperature (blue), and quite large difference
in IR12.0 and IR10.8 channels (red). In contrast, lofted dust plumes with fine quartz particles have
similar values of emissivity at IR10.8 and IR8.7 and this makes dust appear pink or magenta. Deep
cumulonimbus clouds are depicted with red colors, while thick water clouds appear yellow. The RGB
dust product is a very useful tool to qualitatively monitor dust transport events, taking advantage of
the high temporal resolution of SEVIRI observations. The dust RGB product is provided in hourly
intervals by EUMETSAT (European Organization for the Exploitation of Meteorological Satellites)
and is used in this work to monitor the evolution of the dust transport event.
In some cases, however, the usefulness of the product can be limited and this should be considered
in the following discussion. First, the visual contrast of dust and the underlying surface is diminished
when the temperature difference of the two is low, e.g. during nighttime. Second, high levels of
columnar water vapor or the presence of the temperature inversion can mask the presence of dust
in the atmosphere (Brindley et al., 2012). Finally, the contrast of dust and the ground can be further
diminished over some type of surfaces e.g. over rocky terrain, due to its high emissivity at the 8.7
μm channel (Banks and Brindley, 2013).

**2.2 Modeling**
**2.2.1 RAMS-ICLAMS model**
For the simulations used in this study we adopt the online coupled atmospheric and air quality
modeling system RAMS-ICLAMS (Pielke et al., 1992; Meyers et al., 1997; Cotton et al., 2003;
Solomos et al., 2011). The Integrated Community Limited Area Modeling System (ICLAMS) is an
enhanced version of RAMS6.0 and it has been developed by the Atmospheric Modeling and
Weather Forecasting Group at the University of Athens, Greece. The model is set up in a two-way
nesting configuration. The external domain grid space is set at 12×12 km and the grid space of the
inner domain is set at 4×4 km. A higher resolution (cloud resolving) grid at 2×2 km is nested over the
haboob generation area at Syria-Iraq-Iran-Turkey borders. The locations of the model domains are
shown in Figure 1. The vertical structure of the model consists of 50 sigma-z terrain following levels.
The first model level is at 50 m above ground and the levels stretch from the surface up to 18 km.
The dust emission scheme follows the saltation and bombardment approach (Marticorena and
Bergametti 1995; Spyrou et al., 2010). Wet and dry deposition of dust is formulated following
Seinfeld and Pandis 1998. Mineral dust is represented with a transport mode of eight radii bins
namely 0.15, 0.25, 0.45, 0.78, 1.3, 2.2, 3.8 and 7.1 μm. Sea salt aerosol is also parameterized
following Monahan et al., 1986; Zhang et al., 2005; Leeuw et al., 2000 and Gong et al., 2002, 2003
and it is represented with an accumulated and a coarse mode at 0.18 μm and 1.425 μm in radius
respectively. Dust and sea salt particles interact with the radiative transfer code of the model (Rapid
Radiative Transfer Model (RRTM), Mlawer et al., 1997; Iacono et al., 2000) for the computation of
heating rate fluxes. The formation of cloud condensation nuclei (CCN) and ice nuclei (IN) from dust
and sea salt particles is also included in the model based on the schemes of Fountoukis and Nenes,
2005 and Barahona and Nenes 2009. Initial and boundary conditions are from the NCEP final
analysis dataset (FNL at 1°×1° resolution) and the sea surface temperature is the NCEP operational
SST at 0.5°×0.5°. The convective parameterization scheme of Kain and Fritsch, 1993 (KF) is activated
for the two coarser grids. Assimilation of radiosonde data from the airports of Adana (36.98°N,
35.35°E, 00Z and 12Z), Bet Dagan (32.00°N, 34.81°E, 00Z and 12Z), Diyarbakir (37.54°N, 40.12°E, 00Z
and 12Z), Mafraq (32.36°N, 36.25°E, 21Z) and Nicosia (35.10°N, 33.30°E, 00Z) is also activated to fine
tune the simulations. A series of sensitivity runs with various model configurations (different
physical schemes, assimilation parameters and domain structures) is performed until we conclude
to the optimum setup for the specific simulation.
**2.2.2 Land use changes and activation of dust sources**
An accurate representation of dust sources in the region is crucial for understanding this complex
dust event, but this is hampered by seasonal and interannual variability of dust sources  together
with  recent land cover changes in the region. The original land use database of the model is the
USGS Data Base Version 2 which is obtained from 1-km AVHRR data (Advanced Very High Resolution
Radiometer) spanning April 1992 through March 1993. Firstly, this annual-mean dataset cannot
accurately describe the land use and dust sources at the end of the dry season in the Middle East.,
Secondly, the complex interactions of drier climate (Notaro et al., 2015; Cook et al., 2016),
transboundary water managements (Voss et al., 2013), and prolonged conflict (Jaafar and Woertz,
2016) have led to further changes of land use types that are no longer reflected at the model and
this could have a large impact on dust production. The comparison of Landsat 8 natural color and
NDVI imagery between August 2013 and 2015 (Figure 2) reveals large areas of uncultivated fields in
regions of contested boarders and exposed river and lake-bed sediments especially around the
Euphrates river, all of which are known to be very efficient dust sources (Prospero et al, 2002;
Ginoux et al., 2012).  The impact of the ongoing conflict on land use and vegetation can be further
highlighted in Figure 3, showing the time series of MODIS NDVI in the region around Hawija, Kirkuk
province, Iraq (region B of Fig.2). Agriculture in Hawija is based on a combination of rain-fed and
irrigated fields, in accordance with a rainy and a dry season from November to May and from June
to October respectively. The NDVI time series clearly captures this behavior, with a major annual
NDVI pick during wet months and a smaller cycle during each summer, probably reflecting the
growth of summer crops with the help of irrigation. This summer cycle is completely absent in 2015.
Indeed, a recent survey of the Food and Agriculture Organization (FAO) of the United Nations,
reveals that large parts of the irrigation system in Kirkuk and surrounding regions have been
destroyed by military operations and a large number of pumps and generators required for
irrigation have been stolen (Singh N. et al, 2016). This, together with the destruction of other
agricultural equipment and infrastructures, has severely disrupted the summer agriculture activities
of 2015, exactly before the dust storm studied here, leaving the fields to act as very efficient dust
sources.
In order to get most accurate representation of dust sources for the specific event we use 1km
monthly Normalized Difference Vegetation Index (NDVI) from MODIS (Didan K., 2015) to
characterize land use type in the region of interest. Specifically, we consider regions with NDVI
values from 0 to 0.1 to correspond to bare soil and consequently efficient dust sources (DeFries et
al., 1994). The updated land cover dataset is used for all results shown in this study. Results from
simulations using the older database are only shown in Figure 11 for comparison.


**3. Results**

**3.1 Meteorological conditions**

The main driving force for the generation of this extreme dust episode is the combination of two distinct meteorological features in the greater area: (i) establishment of a thermal low over the bare-soil areas of Syria and (ii) convective outflow boundaries at the mountains of Iraq and Syria. These processes are analyzed in the following sections using modeling results and remote sensing.

**3.1.1 Development of a low pressure system over Syria on 6 September 2015**

As seen at the outer model grid in Figure 4a, the passage of a trough is evident over Turkey on 6 September 2015, 00:00 UTC. The low pressure center at 500 mb is found at 5840 m over the east bank of Black Sea. During the same day, radiative warming of the bare soil surface results in very hot soil temperatures exceeding 50 °C in Syria and Iraq. Advection of warm air from the Red Sea is also evident at the lowest troposphere by the 1000-700 mb thickness in Figure 4b. This combination of cold air aloft with low level warming, leads in the formation of a thermal low pressure system over Syria that is evident by the 925 mb geopotential height in Figure 4c. Another parameter that plays important role for the process of dust source activation is the difference between surface temperature ($T_{Surf}$) and air temperature at 2m ($T_{2m}$). Findings from earlier field experiments (i.e. SAMUM) show that such a difference of 17°C-20°C facilitates the uplift of convective dust plumes (Ansmann et al., 2009). As seen in Figure 1, the modeled $T_{Surf}$-$T_{2m}$ difference at 10:00 UTC exceeds 17°C over extended bare soil areas in Syria. This temperature gradient further explains the effectiveness of dust production at these areas. The pressure system and the associated cyclonic flow persist during the entire day of 6 September 2015 and result in the mobilization of dust in the area. Dust uptake is mostly evident at the outer parts of the cyclone where surface wind speed exceeds 7 m s$^{-1}$ almost during the entire day and $T_{Surf}$-$T_{2m}$ obtains maximum values. The elevated particles are quickly distributed inside the system and a distinct cylindrical dust cloud is soon formed. Recirculation of the elevated dust particles inside the closed cyclonic flow results in extreme AOT values exceeding 15 at specific areas as seen in Figure 4c. The formation of this dense dust plume is also evident in the MSG-SEVIRI satellite dust RGB image in Figure 4d. Pink and purple colors in this image indicate dust while brown and red colors indicate clouds. The convective outflows from the Zagros Mountains in Turkey that are evident by the black dashed line and wind

vectors at 925 mb in Figure 4c enhance the mobilization of dust at the northern parts of the heat
low. Transport of dust from Lebanon towards Cyprus  is evident at the satellite and modeling
images on 7 September 2015, 00:00 UTC (Figure 5).  This cut-off plume (plume_1) travels in the
lower troposphere above the marine boundary layer and it was observed at 1.5 km above Limassol
on 7 September, 19:00 UTC as reported by Mamouri et al. (2016). The faster propagating haboob
plume (plume_2) was detected at 2.0-3.5 km over Cyprus at 19:00 UTC. The extreme AOT values
(>10) that are seen in Figure 5a over Syria result from the overlapping of cyclone-driven and
haboob-driven dust over this area. In the model, approach of the haboob front in Syria is
accompanied also by cloud formation as seen by the 70% cloud-cover contours in Figure 5a;
however these clouds are not evident in the satellite image (Figure 5b). The more elevated (cyclone-
driven) dust in Figure 5b is shown in pink (plume_1 over the sea and plume_3 over Northern Syria
and Southern Turkey) and the near surface dust (haboob) is shown with a darker purple color
(plume_2).

**3.1.2. Convection and haboob generation on 6 and 7 September 2015**
At 13:00 UTC on 6 of September a northward low level flow is evident over N. Iraq and N. Syria
(Figure 6a). This relatively unstable air mass is characterized by increased equivalent potential
temperature (theta_e) reaching up to 508 K. This flow is associated with a westerly shift of the
Somalian Low Level Jet (SLLJ). The SLLJ is part of the West India Monsoon circulation and is shown in
Figure 6b. It is characterized by strong SW winds blowing from the Somali highlands towards West
India. This low level flow steers towards the west along the coastal mountains of Yemen and Oman
and results in SE winds transferring moisture from the Arabian Sea towards the inlands of the
Arabian Peninsula. Low-level advection of warm air masses from the Red Sea and Saudi Arabia
towards the storm area is also evident in Figure 6c by the 1000-700 mb modeled thickness at 15:00
UTC. Mechanical elevation of this relatively unstable air as it approaches Mt. Sinjar in N. Iraq
triggers convection at this area. A number of atmospheric parameters that determine the formation
of the cold pool are shown in Figures7a-d. As seen in Figure 7a, the iso-temperature line of -20°C
between rain droplets and ambient air temperature clearly defines the cold pool area. Sub-
saturated air below the cloud base is also evident in Figure 7a since the relative humidity at the
neighbor of the convective cloud is between 15-20 %. The combination of sub-saturated air and
temperature gradient results in a faster evaporation rate of the rain droplets and in the formation
of a cold pool at the area of Northern Iraq with speeds ranging from 10 up to 20 ms$^{-1}$ (Figure 7b). As
seen in Figure 7c, the convective cloud top reaches 12 km and the updrafts exceed 18 m s$^{-1}$ at 15:00
UTC. The rainfall curtain (downdraft area in Figure 7c) extends up to 4-5 km and the severity of the
storm leads in the formation of a haboob that is evident by the streamline structure and dust
production below the non-precipitating parts of the cloud in Figure 7d. A Kelvin-Helmholtz billow at
2-3 km separates the density current head from the ambient flow similar to previous findings for
convective haboobs (Solomos et al., 2012).Turbulence distributes the dust particles inside the
system and dust concentrations exceed 2000 µg m$^{-3}$ close to the surface. As the cold pool moves
towards the North it triggers the generation of secondary convective cells at the mountainous areas
along Iran-Iraq-Turkey borderline. At 20:00 UTC, a series of convective outflows converges in an
organized SE propagating density current that is evident in the model over N. Iraq and N. Syria
(Figure 8a). This system is characterized by wind speeds higher than 6 m s$^{-1}$ and results in activation
of dust sources and near surface modelled concentrations largely exceeding 10000 µg m$^{-3}$ (Figure
8b). However, the corresponding SEVIRI image (Figure 8c) indicates that by this time the haboob has
already penetrated about 200 km inside Syria which is not reproduced by the model. The latency
between satellite and modeled haboob fronts is possibly attributed to a slower propagating
modeled haboob or to a triggering delay of convection in the model due to the imperfect
representation of boundary layer properties and atmospheric stability.

**3.2 Dust cloud properties and comparison with observations**
**3.2.1. Vertical dust structure**
The dust layer structure as it propagates towards the Mediterranean is captured by two CALIPSO
overpasses at 23:33 UTC, 6 September 2015 (Figure 9) and at 10:35 UTC, 7 September 2015 (Figure
10). Collocated model cross sections of dust and MSG-SEVIRI dust images are also presented in
Figures 9 and 10. All heights in satellite and model profiles refer to heights above surface. At the 6
September overpass, the southern part of the dust layer (31°N-34°N) is detected up to 2-3 km and
originates from the cyclonic flow over Syria (Figure 9c). The modeled dustload is also shown in
Figure 9b for comparison. Dust concentrations are estimated from CALIOP lidar backscatter signals
(see Section 2.1.2) and as seen in Figure 9c they reach up to 5000 µg m$^{-3}$ close to the surface
between 31°N-34°N and higher than 6000 µg m$^{-3}$ in the first 500 m. Similar structure and dust
concentrations are also found by the model (Figure 9d). The northern part of the overpass (35°N-
38°N) detects also elevated dust due to cyclonic activity between 2.5-4.5 km and concentrations up
to 1000-2000 µg m$^{-3}$ are evident at this area from CALIPSO. The model overpredicts dust at this area
with simulated concentrations reaching up to 5000 µg m$^{-3}$. These elevated layers are shown with
pink colors in Figure 9a. Low level dust (purple colors in SEVIRI images) is also evident in this area
due to the propagating haboob and CALIPSO detects this two-layer structure with a clear separation
at 2 km. The model also reproduces the uplift of dust at 35°N where the two systems (cyclone and
haboob) merge. The modeled concentration inside the haboob reaches extraordinary values
exceeding 10000 µg m$^{-3}$. Due to the severity of the event, the CALIPSO lidar signal is totally
attenuated below ~1 km (dark blue color), in the area between 35°N-37°N. For that reason the
information from the satellite is limited at this area in the top 500m of the propagated haboob (1-
1.5 km), implying also the existence of much higher values close to the surface.
The second overpass at 7 September 10:35 UTC is actually behind or at the tail of the propagating
dust storm (Figure 10a). The modeled dustload is also shown in Figure 10b for comparison. The thin
dust layer that is detected by CALIPSO between 30°N-32°N reaches up to 2 km and maximum dust
concentrations of up to 2000-3000 µg m$^{-3}$ are calculated mostly close to the surface (Figure 10c).
Extreme dust concentrations are also found in both satellite (Figure 10c) and model plots (Figure
10d) between 34°N-36°N at the tail of the propagating system. Dust values at this area are so high
that CALIPSO observation again suffers from total attenuation of the lidar signal after penetrating
the first 1000 m and extraordinary concentrations of up to 20000 µg m$^{-3}$ are found in the lower
model levels (up to 1.5 km). Similar values are observed from CALIPSO at the edge of the haboob
(33.5°N-34°N) where the signal is strong enough to provide valuable information. The elevated
layers (2-4 km) between 36°N-38°N at both CALIPSO and model profiles are dust residuals over the
mountains of Turkey. An elevated dust layer of up to 600 µg m$^{-3}$ is also found south of 35°N in the
model at heights between 3-4 km. Due to the aforementioned latency between the true and
modeled propagation speeds, the model cross-section is closer to the core of the system hence this
layer consists of modeled cyclone uplifted dust that in fact is already west of the CALIPSO ground
track.

**3.2.2. Dust load over Cyprus**
The observed structure and amounts of dust arriving in Cyprus is described in detail by Mamouri et
al. (2016). The arrival of the dust plumes at Limassol in Cyprus is evident in Figure 11. A double layer
structure is detected by the lidar on 7 September 19:00 UTC. The relatively shallow dust layer that is
found between 0.8-1.7 km with a maximum peak at 2000 µg m$^{-3}$, comes from the detached dust air
mass traveling off the coast of Lebanon as described in Section 3.1.1. The model reproduces
correctly the height of this layer but the maximum concentration is underestimated by almost 50 %.
The upper layer that is detected between 1.8-3.6 km originates from the north part of the fast
propagating haboob that catches up with the "Lebanon" dust over Cyprus. The location and dust
concentrations of this layer are adequately reproduced by the model. The total modeled dust load
is similar to the observed (lidar) dust load but the vertical distribution of dust in two distinct layers
is not so clearly reproduced. Model results using the old vegetation database are also shown in
Figure 11. As seen by the dashed line in this plot, this simulation failed to reproduce the strength of
the event and the maximum concentration is 400 µg m$^{-3}$ at about 0.7 km height. On 8 September
the lidar system could not operate due to the extraordinary dust load. The mean MODIS derived
AOT on this day varied between 1.5-5 over five sites in Cyprus (Pafos, Limassol, Larnaca, Nicosia,
Rizokarpaso), (Mamouri et al., 2016). Given the fact that the maximum retrievable MODIS AOT is 5,
these values are most probably an underestimation of the true AOT. The distribution of modeled
AOT during 00:00 UTC-15:00 UTC on 8 September is shown in Figure12; the dust plume approaches
Cyprus from the South and the orographic effect of Mt. Troodos results in an inhomogeneous
distribution of dustload over the island, which explains the AOT variability between the sites. The
modeled AOT values over the Middle East inland exceed 10 as shown also by the sharp gradient
towards the eastern part of Figure 12 plots. However the extreme dust storm affecting Cyprus
during 8 September is the result of a plume that approaches the island from the south. This dust
layer is evident between 1.5-3.5 km in the vertical cross-sections of model dust concentration at
00:00 UTC and 03:00 UTC in Figure 13. Differential heating between the land and water bodies and
between flat terrain and mountain slopes results in the development of local wind flows
(downslope / upslope winds). Downward mixing of dust as this air mass approaches the topographic
barrier of Troodos mountain increases the near-surface concentrations at the southern sites
especially during local morning and noon hours (06:00 UTC, 09:00 UTC). In the afternoon hours
(12:00 UTC, 15:00 UTC), the development of upward motions over Mt. Troodos separates the dust
flow over Cyprus into two distinct cells (a south and a north one) and at this time increased
concentrations are found over the northern sites of the island. The maximum simulated
concentrations are up to 4000 µg m$^{-3}$ aloft and about 1000 µg m$^{-3}$ close to the surface.
Taking into account the complexity of the situation, the spatiotemporal evolution of the episode
seems to be correctly explained by the model but the extreme values of 8000-10000 mg m$^{-3}$ that
are reported by Mamouri et al. (2016) are not reproduced. This discrepancy can be attributed to a
variety of reasons related to both dust and atmospheric properties that are not properly resolved at
this model scale (e.g. more intense downward mixing, increased emissions from the sources,
limitations due to emission size bins, inaccurate deposition rates etc.). The simulated versus
observed maximum AOT values for the five sites are also shown in Table 2. The model reproduces
the higher AOTs at the most southern sites (Limassol and Pafos) compared to the central and north
sites. Following the previous discussion about the already underestimated MODIS AOT it seems that
the model reproduces the distribution of dust over Cyprus however with an overall underestimation
of more than 2. A possible explanation could be also that the dry river beds of Tigris and Euphrates
as well as several dust sources over Syria and North Iraq provide even more erodible sediments
than those assumed by the model hence the discrepancies in dust concentrations.

**4. Discussion and Conclusions**
A combination of meteorological and land-use conditions resulted in the formation of an
unprecedented dust episode over Middle East and the Eastern Mediterranean during 6-11
September 2015. This event is unique due to the coincidence of various atmospheric phenomena
related with the generation of turbulence and dust production. Interpretation and analysis of
remote sensing data (EARLINET, CALIPSO, MSG-SEVIRI) and modeling simulations (RAMS-ICLAMS)
reveals the main reasons that led in the uplift and persistence of the dust layers.
The major processes affecting the generation of the dust storm are found to be:

1.  The formation of a strong thermal low and of convective outflows over Syria that lifted dust
up to 4 km.
2.  The intrusion of moist and unstable air masses from the Arabian Sea and the Red Sea that
triggered convective activity over Iraq-Iran-Syria-Turkey borderline.
3.  The generated outflow boundaries that led in dust deflation and formed a westward
propagating haboob merging with the previously elevated dust over Syria.
4.  The increased efficiency of Middle East dust sources as an aftermath of war and the related
changes in land use.

As reported by Mamouri et al. (2016), almost all operational dust models failed to forecast this
event. RAMS-ICLAMS in this study is not used in forecasting mode but rather as a tool for the a-
posteriori analysis and explanation of the event. This means that the configuration of several model
parameters such as the nested grid structure, convective parameterization schemes, dust source
strength etc. is guided by the available observations. In this context, most observed processes are
successfully described by the model and the various physical mechanisms that took place during the
event are explained. However, certain inaccuracies in the quantification of atmospheric variables
and spatiotemporal deviations in the description of convection and other physical processes can still
significantly decrease the model skill especially regarding the quantification of dust mass profiles.
The analysis presented here raises considerations regarding the forecast skill of the atmospheric
dust models, since even though such extreme episodes are seldom they still represent the most
threatening dust hazards. The long range transport and the general circulation of dust in the
atmosphere are now often adequately forecasted by most global models but convectively driven
episodes cannot be resolved at synoptic and mesoscale resolutions. Moreover, a recent study by
Pope et al. (2016) at the area of Sahel / southern Sahara suggests that unresolved haboobs during
the summer monsoon may be responsible for up to 30% of the total atmospheric dust and such
considerations raise questions on the current status of early warning systems for dust episodes. It is
probably obvious that such a system cannot rely exclusively on modeling simulations. As shown at
the present study, the complexity of these events makes forecasting them very challenging and it is
possible that a certain model configuration could successfully reproduce a specific event but not all
similar events. The key for forecasting these events in atmospheric models is the use of cloud
resolving grid space. However, such high resolution grid-space can only be applied over limited
areas due to restrictions in computational power. Forthcoming studies using an extended cloud-
resolving grid over the entire Middle-East (e.g. Gasch et al., 2017) could provide more detail on the
individual atmospheric processes during this episode.
Remote sensing can play an important role for the provision of more accurate dust forecasts.
Engagement of geostationary satellite observations (MSG, Sentinel-4) and CALIPSO profiles in
forecasting activities could improve the forecasting skill either by the direct assimilation of satellite
data in dust models or by issuing human-assisted early warnings. Expansion of a lidar network close
to dust source areas (e.g. Sahara, Middle East) will also complement model activities through the
provision of ground truth observations for the vertical profile of dust plumes.  Additionally, the
activation of correlated observations from the EARLINET network following a dust forecast notice
will allow a closer investigation of the physical processes that drive these events.

**Acknowledgements**
The authors acknowledge support through the following projects and research programs: BEYOND
under grant agreement no. 316210 of the European Union Seventh Framework Programme FP7-
REGPOT-2012-2013-1. ACTRIS-2 under grant agreement no. 654109 of the European Union's
Horizon 2020 research and innovation programme. ECARS under grant agreement No 602014 from
the European Union's Horizon 2020 Research and Innovation programme. MarcoPolo under grant
agreement no. 606953 of the European Union Seventh Framework Programme FP7/2007-2013. The
authors acknowledge EARLINET for providing aerosol lidar profiles available under the World Data
Center for Climate (WDCC) (The EARLINET publishing group 2000-2010, 2014a. CALIPSO data were
obtained from the ICARE Data Center (http://www.icare.univ-lille1.fr/) and from the NASA Langley
Research Center Atmospheric Science Data Center. CALIPSO data were provided by NASA. We thank
the ICARE Data and Services Center for providing access to the data used in this study and their
computational center.

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

Table 1. Range and gamma correction for the Red, Green, and Blue channels for construct the Dust
RGB product.

| Color | SEVIRI Channels | Min (K) | Max (K) | Γ |
|-------|-----------------|---------|---------|---|
| Red | IR12.0 – IR10.8 | -4 | 2 | 1 |
| Green | IR10.8 – IR8.7 | 0 | 15 | 2.5 |
| Blue | IR10.8 | 261 | 289 | 1 |



Table 2.  Maximum MODIS and RAMS AOT over Cyprus (8 September 2015)

| | Pafos | Limassol | Larnaca | Nicosia | Rizokarpaso |
|---|-------|----------|---------|---------|-------------|
| MODIS$_{AOT}$ | 3.5 | 5.0 | 5.0 | 2.0 | 5.0 |
| RAMS$_{AOT}$ | 3.5 | 4.0 | 3.0 | 3.0 | 3.0 |





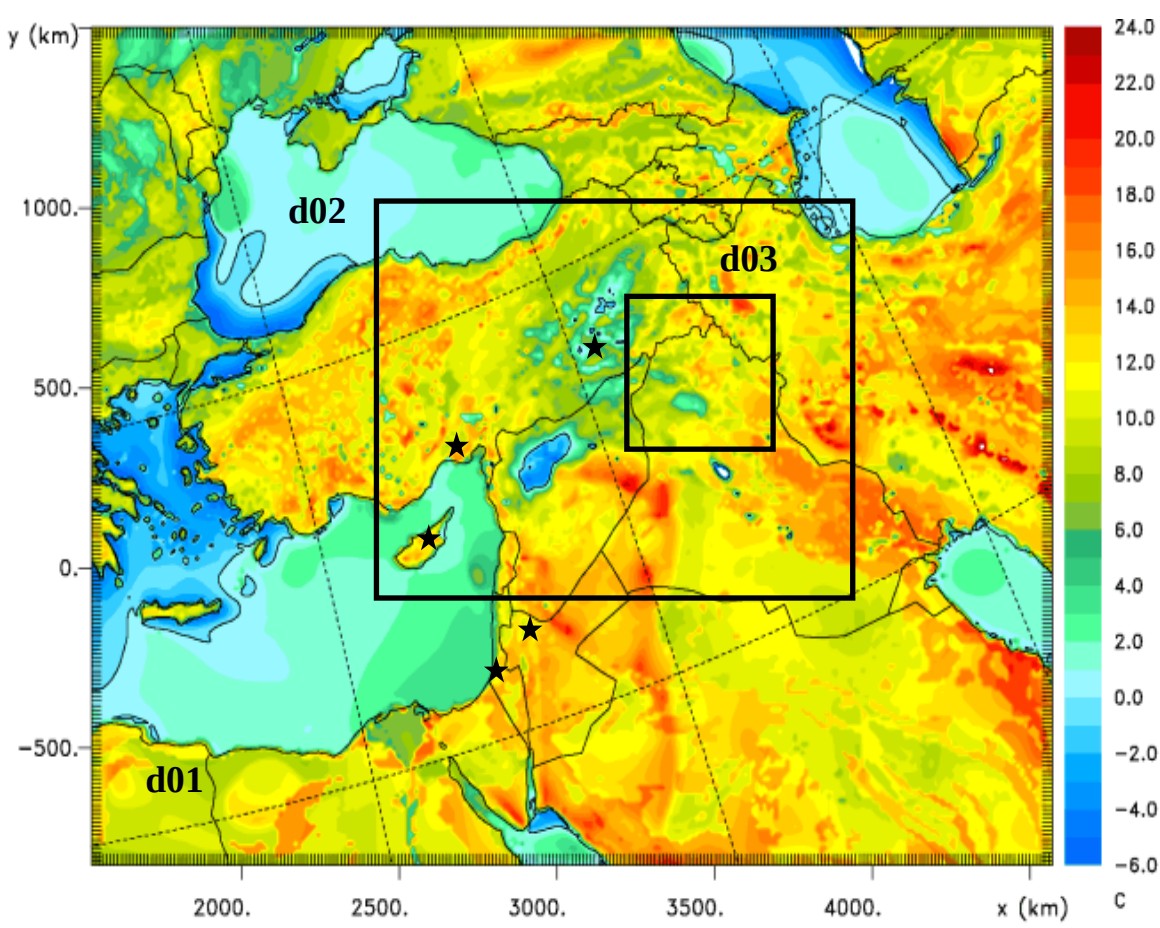


Figure 1. Modeling domain structure and difference (°C) between model soil temperature and model temperature at 2m, 10:00 UTC, 6 September 2015. Black rectangulars indicate the location of the nested model domains (d01:12×12 km, d02:4×4 km, d03:2×2 km) and black stars the location of radiosondes.

631

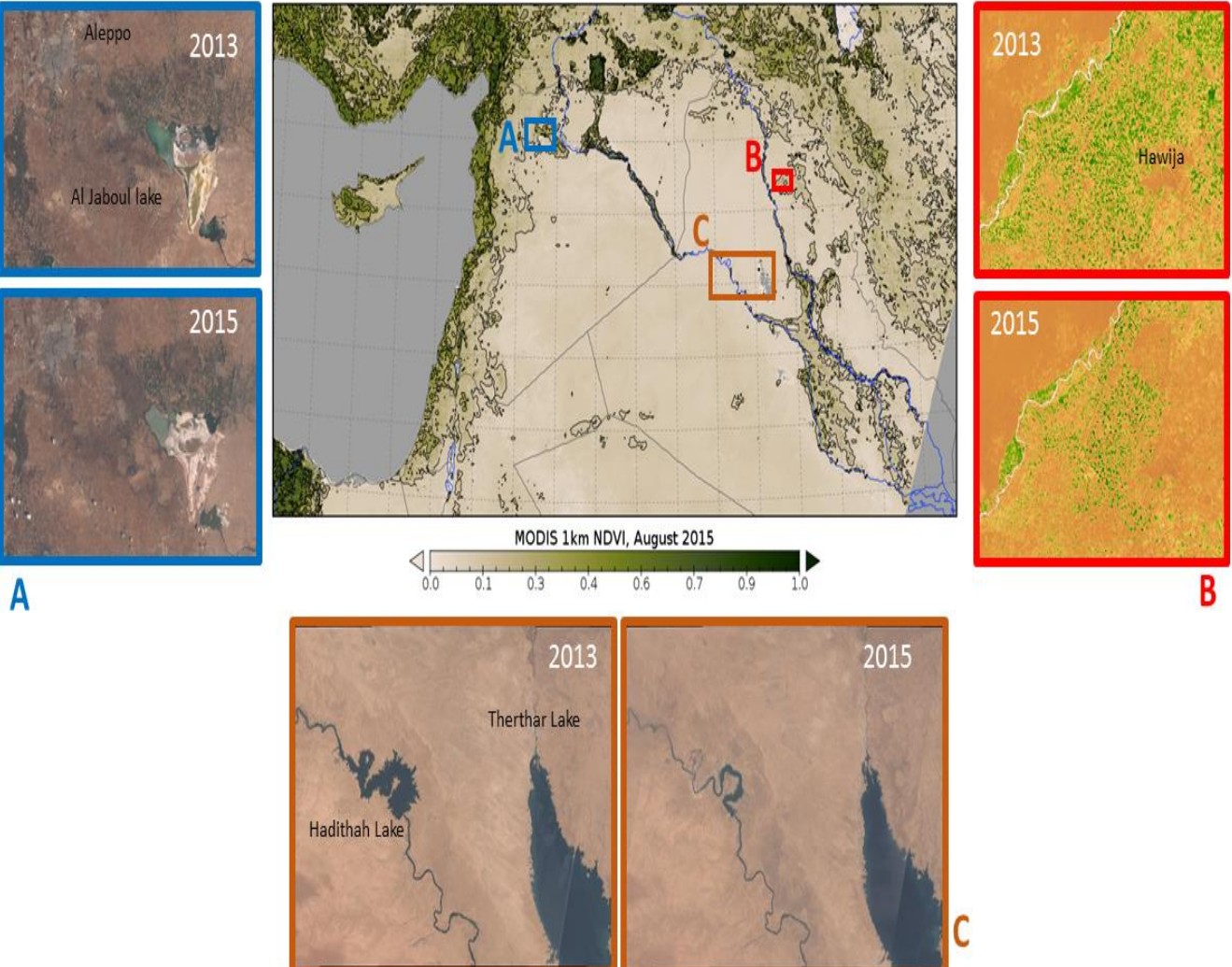

632

Figure 2. (central panel) MODIS NDVI observations for August 2015 were used to identify regions of bare soil that can be sources of dust aerosols. The contour lines correspond to the major ticks of the color scale. Large regions of wester Syria and Iraq have NDVI values from 0 to 0.1. The three subpanels show examples of land type change between summer 2013 and summer 2015. (Subpanel A) Landsat 8 natural color images of Aleppo region, Syria shows changes of cultivation patterns and drying of nearby Al Jaboul lake (e.g. the bright areas of the Al Jaboul Lake - dry parts of the lake - increased from 2013 to 2015); (Subpanel B) Landsat 8 NDVI index images in the region of Hawija, Kirkuk Province, Iraq reveal that large areas remained uncultivated in 2015 (e.g. the 2013 map shows many more green spots - agriculturally used areas - than the 2015 map); (Subpanel C) Landsat 8 natural color images showing diminishing area of Haditah Lake on the Euphrates river and the drying up of the Therthar canal and lake.

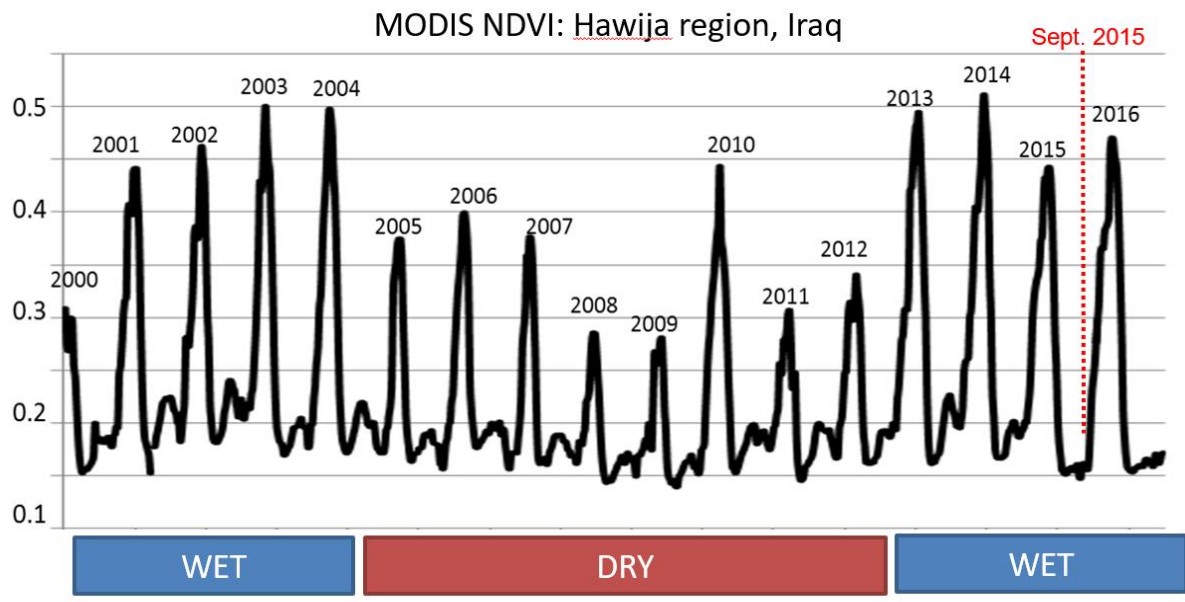


Figure 3. Time series of 16-day NDVI for the agricultural region around Hawija, Kirkuk province, Iraq.
The vertical dashed lined marks the time of the studied dust storm. Note the absence of NDVI
variation in the summer of 2015, prior to the dust storm.

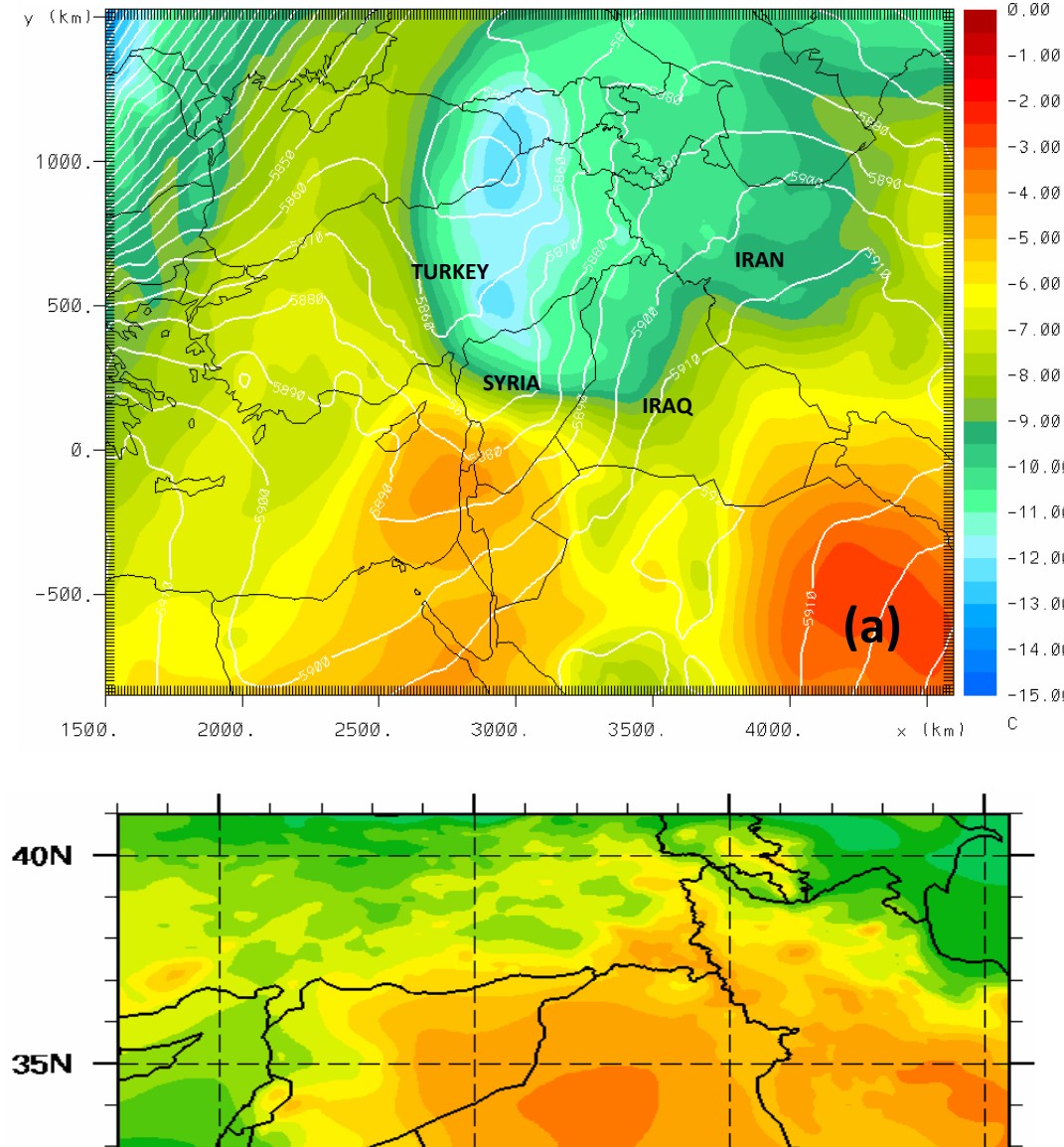



Figure 4. a) Model geopotential height contours (every 10 m) and temperature (color scale in °C) at 500mb, 6 September 2015, 00:00 UTC; b) Model 1000-700 mb thickness (dam), 6 September 2015, 00:00 UTC


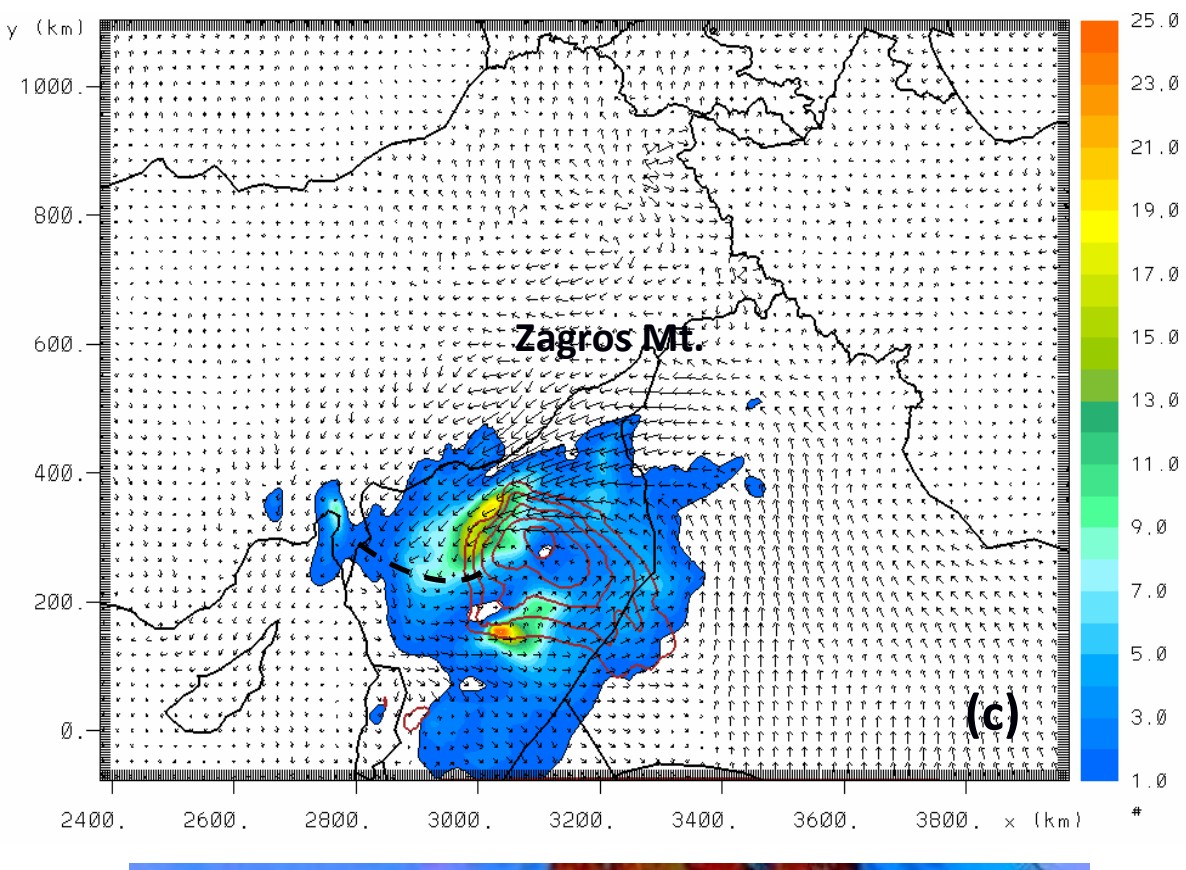


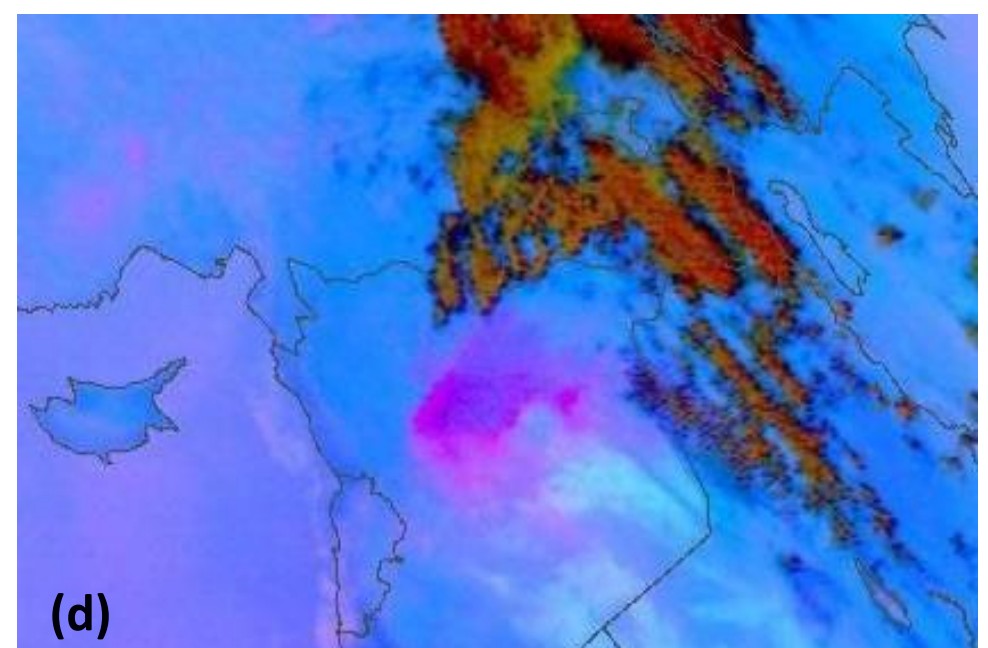

Figure 4.  c) Model AOT (color scale), geopotential height at 925 mb (red contours from 740 to 757.5
m every 2.5 m) and wind vectors at 925 mb. The dashed line denotes the outflow boundaries from
Zagros Mountains; d) MSG SEVIRI dust RGB at 08:00 UTC, 6 September 2015

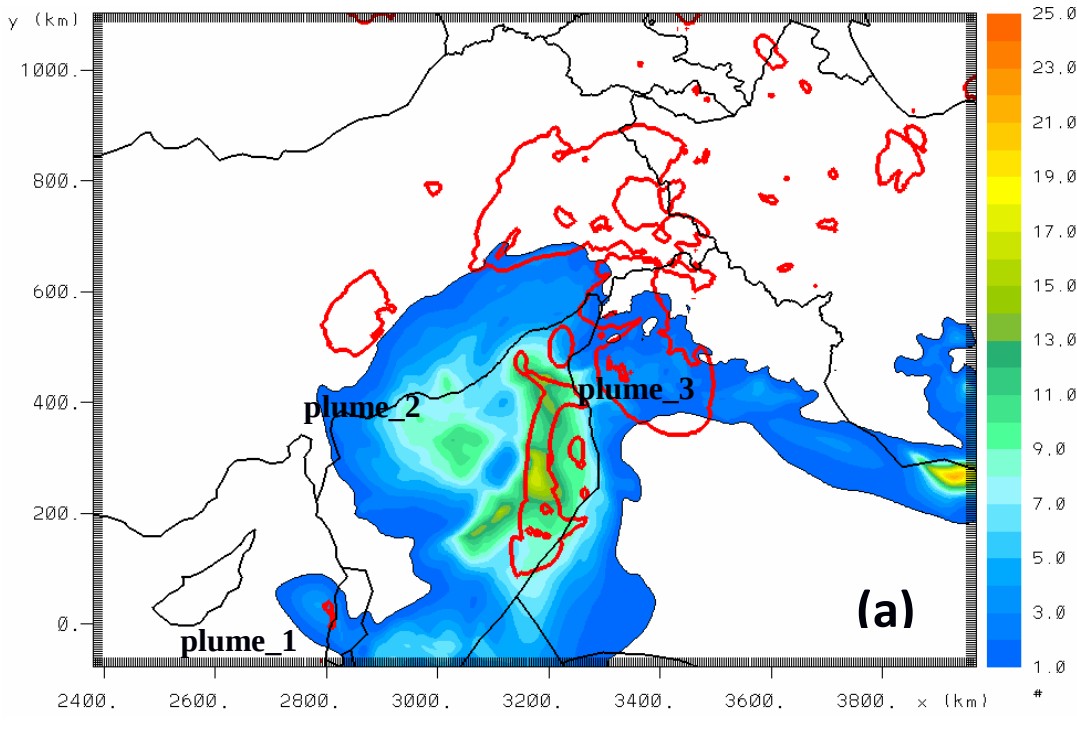

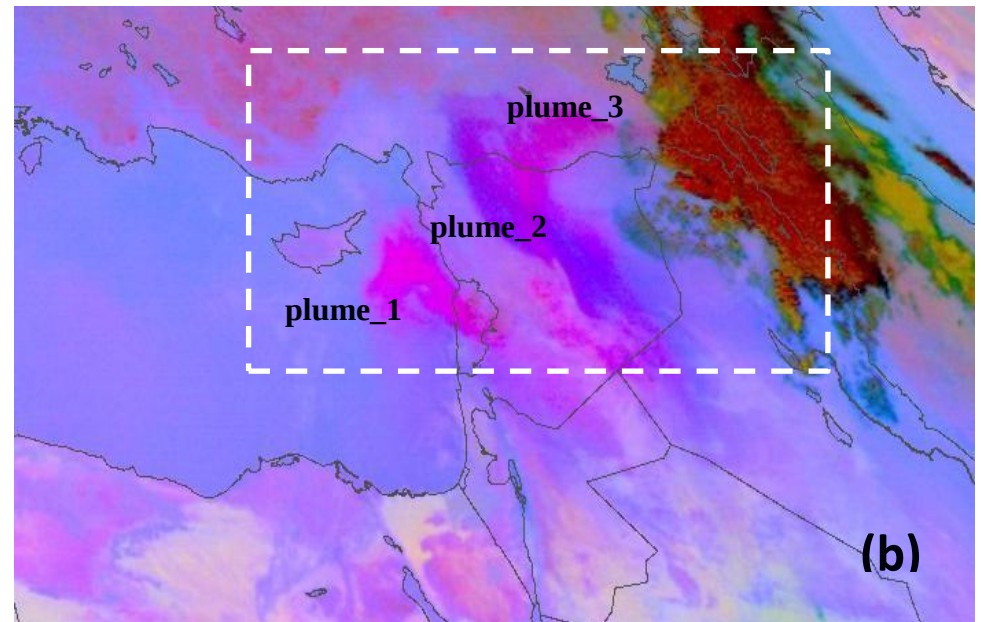

Figure 5. a) Model AOT at 550 nm (color scale) and cloud cover > 70% (red contour). b) MSG-SEVIRI dust RGB component, 7 September 2015, 00:00 UTC. The white rectangular approximately indicates the location of the model domain shown in Figure 5a.



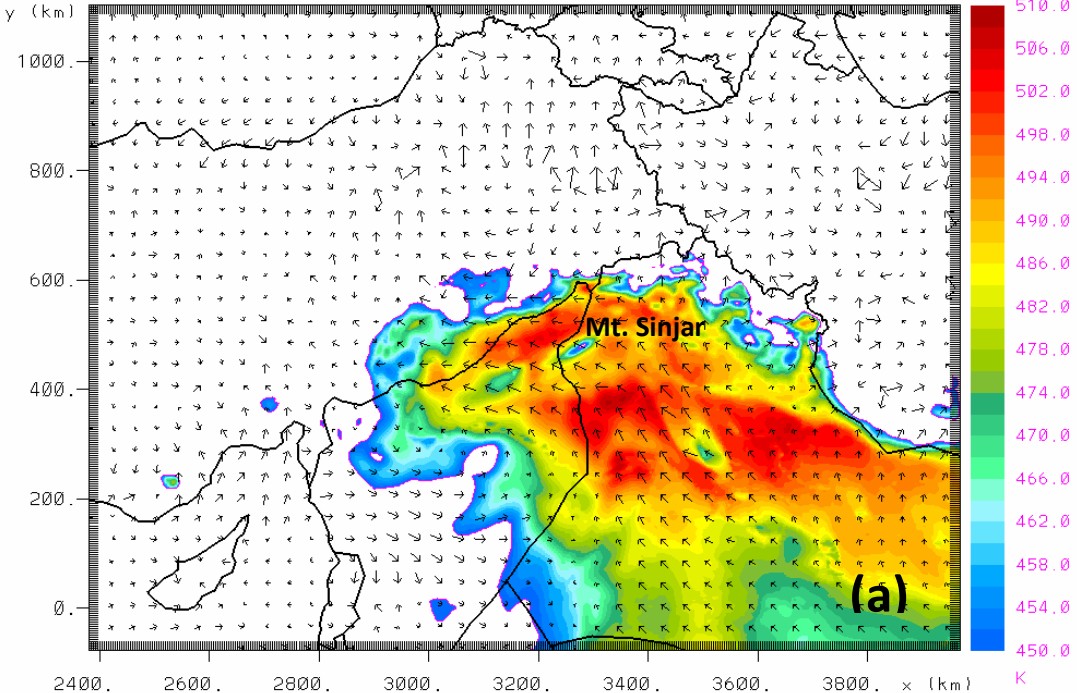

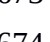

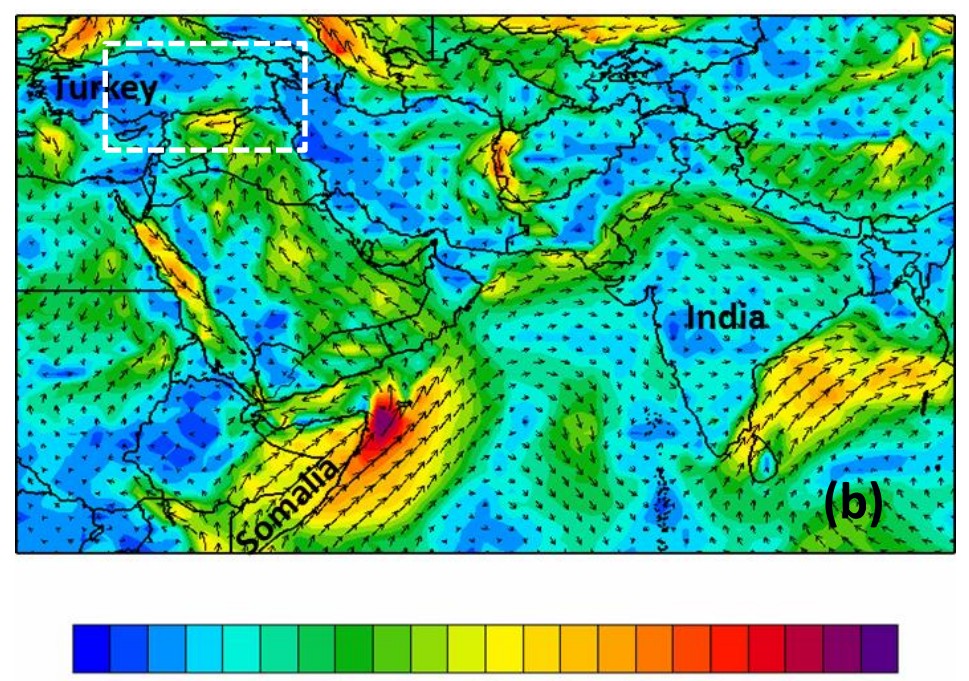


Figure 6. a) Model equivalent potential temperature (K) and wind vectors at 50m above ground, 6
September 2015, 13:00 UTC. b) Wind speed at 975 mb from the NCEP final analysis (FNL) dataset, 6
September 2015, 06:00 UTC. The white rectangular indicates the location of the model domain
shown in Figure 6a.

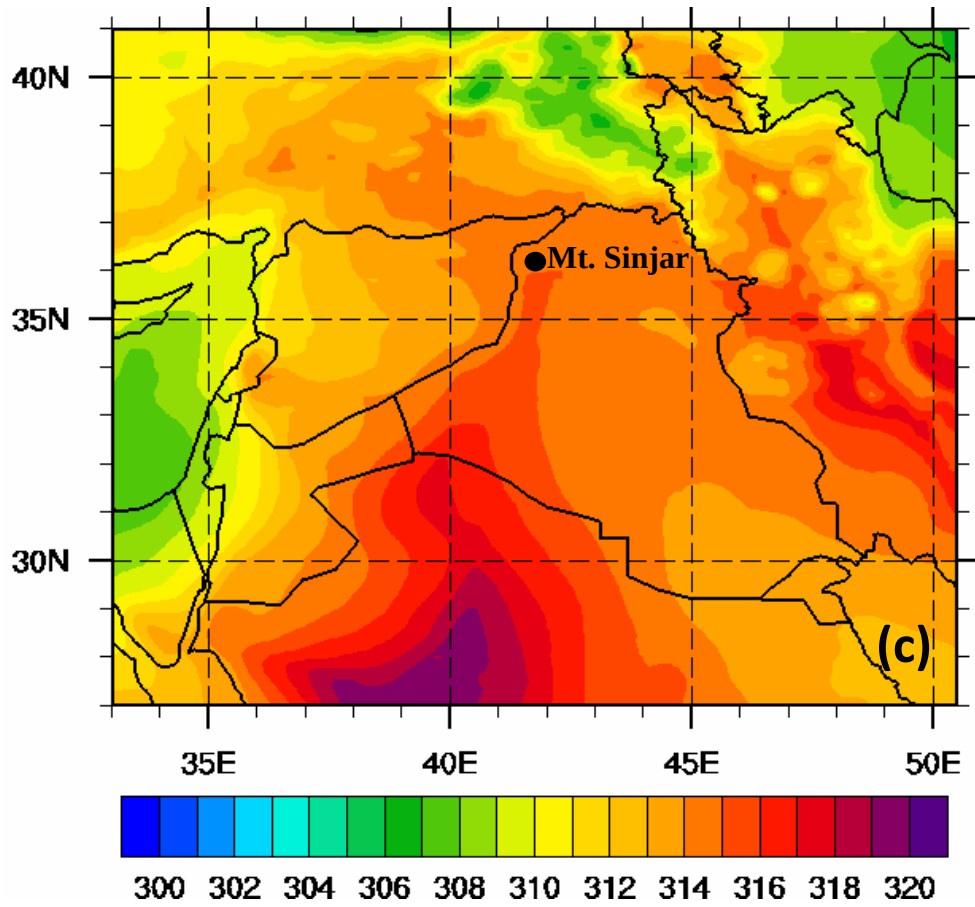


Figure 6. c) Model 1000-700 mb thickness (dam), 6 September 2015, 15:00 UTC




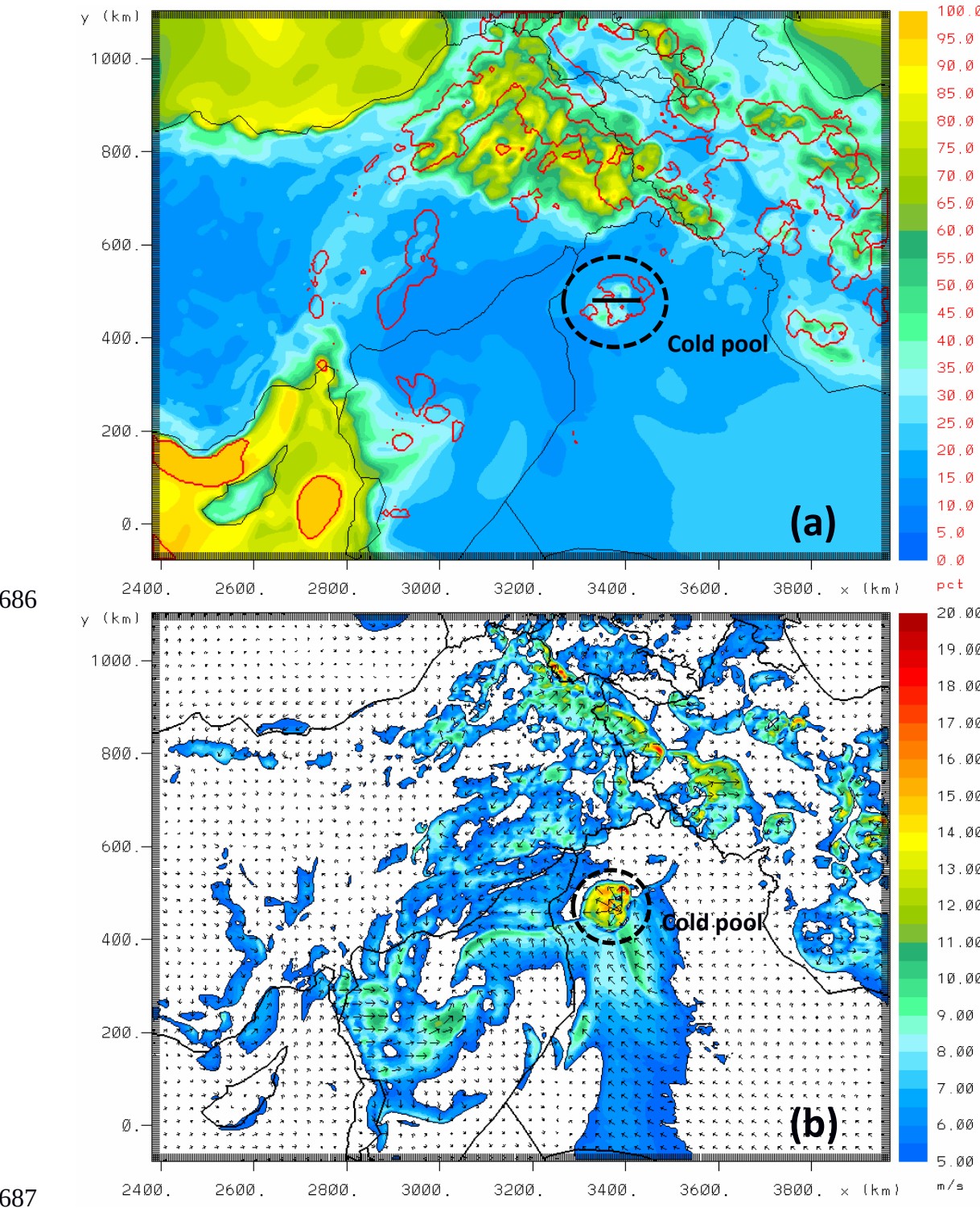



Figure 7. a) Model relative humidity at the first model level (color scale) and -20°C iso-temperature line (red contours) of rain droplets air temperature difference. b) Model wind speed at 10m (ms$^{-1}$). The dashed line denotes the location of the cold pool and the solid black line the location of the storm cross-sections of Figures 7c,d.

692

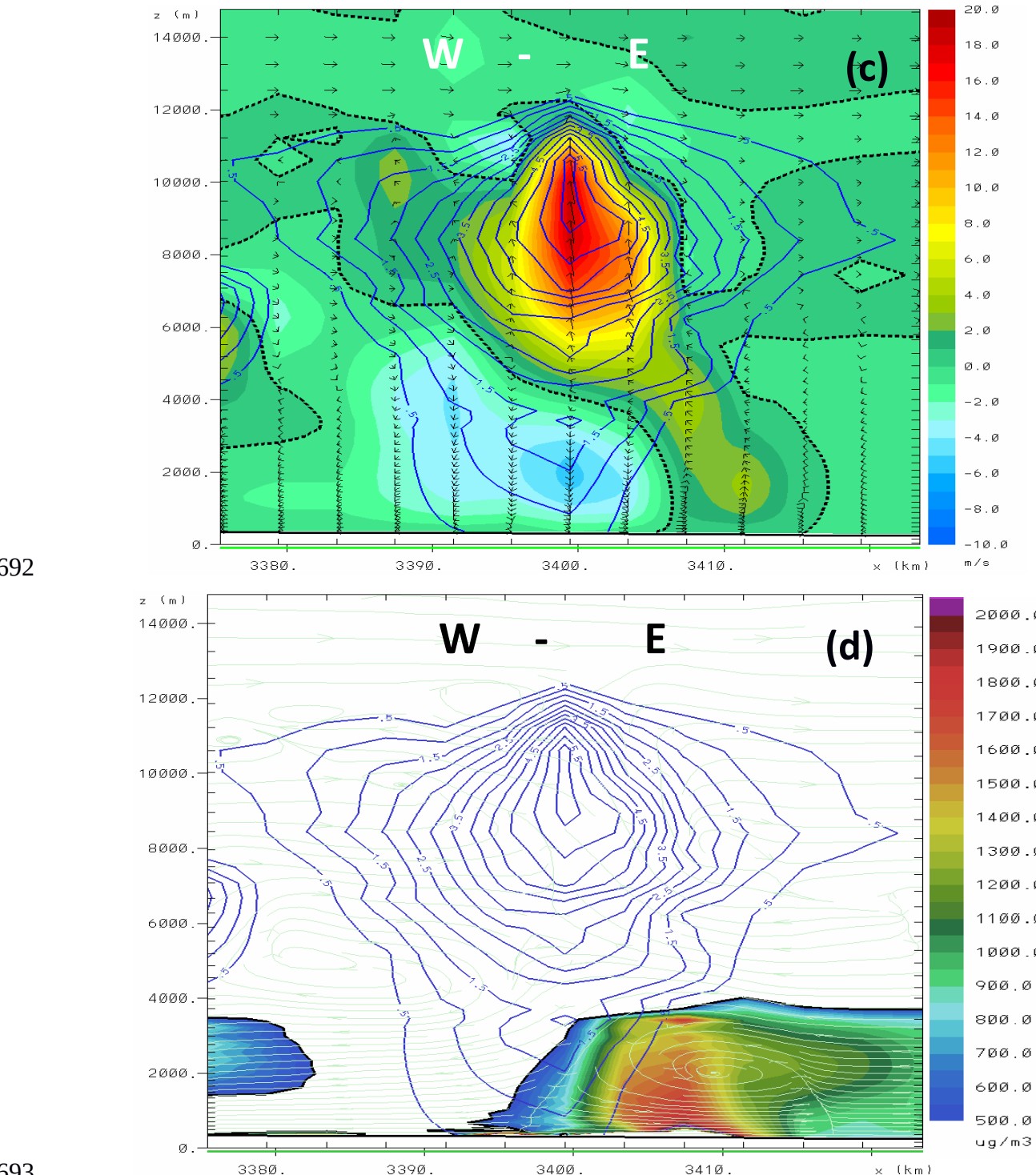

693

Figure 7. c) Vertical cross section of total condensate mixing ratio (blue contours in g k g$^{-1}$) and vertical wind component (vectors and color scale in m s$^{-1}$). The dashed line separates updraft (positive w) from downdraft/precipitating regions (negative w). d) Vertical cross section of total condensate mixing ratio (blue contours in g k g$^{-1}$), dust concentration (μg m$^{-3}$) and flow streamlines, 6 September 2015, 15:00 UTC


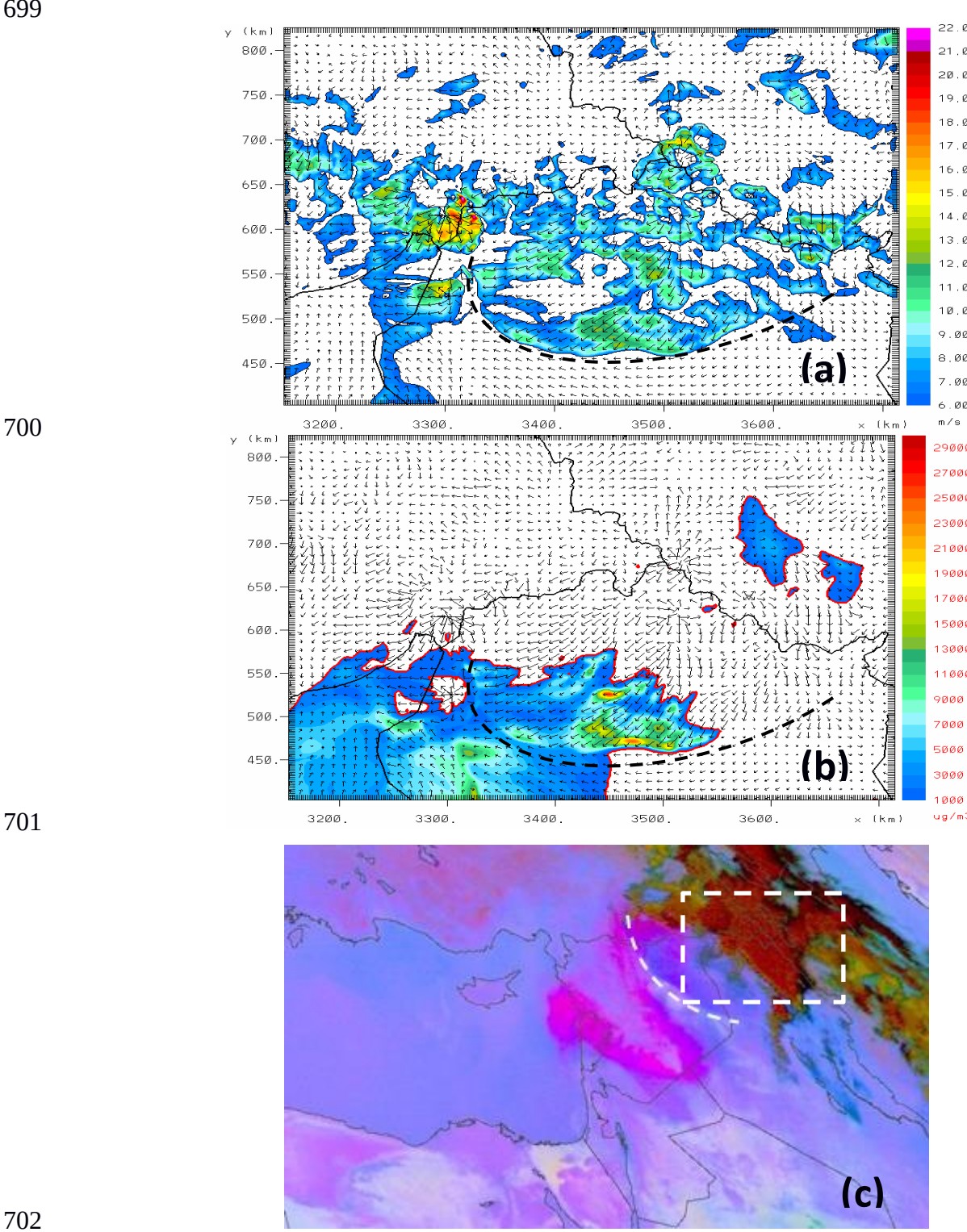




Figure 8. a) Model wind speed greater that 6 ms$^{-1}$ at 10 m and b) Near surface model dust

concentration (µg m$^{-3}$) from the inner grid (2×2km) c) MSG-SEVIRI RGB component, 6 September

2015, 20:00 UTC. The dashed lines indicate the haboob front location and the dashed rectangular in

Figure 8c approximately indicates the location of the model domains shown in Figures 8a,b.


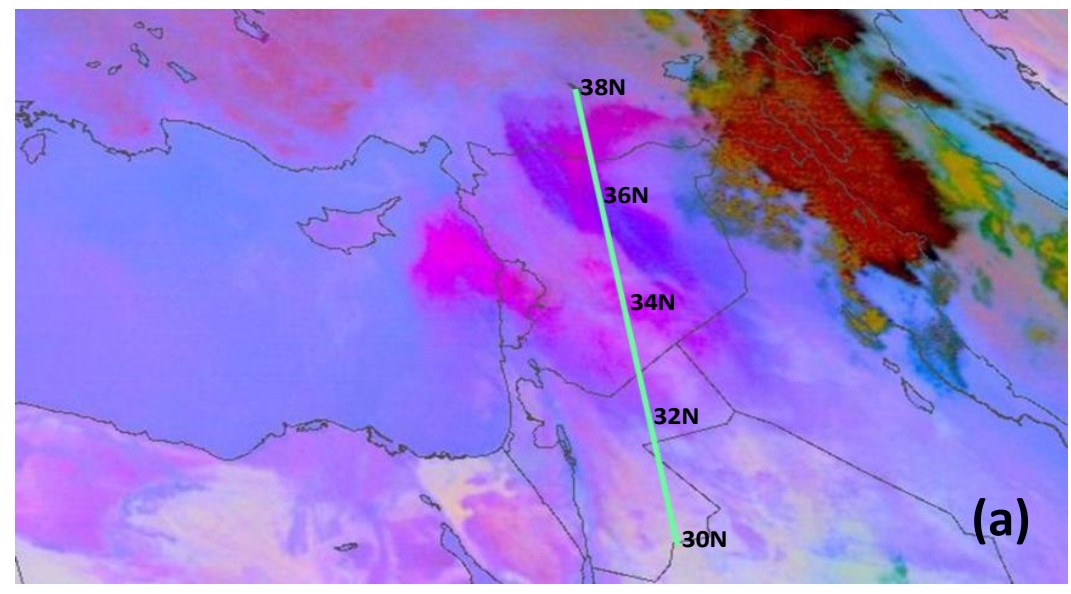


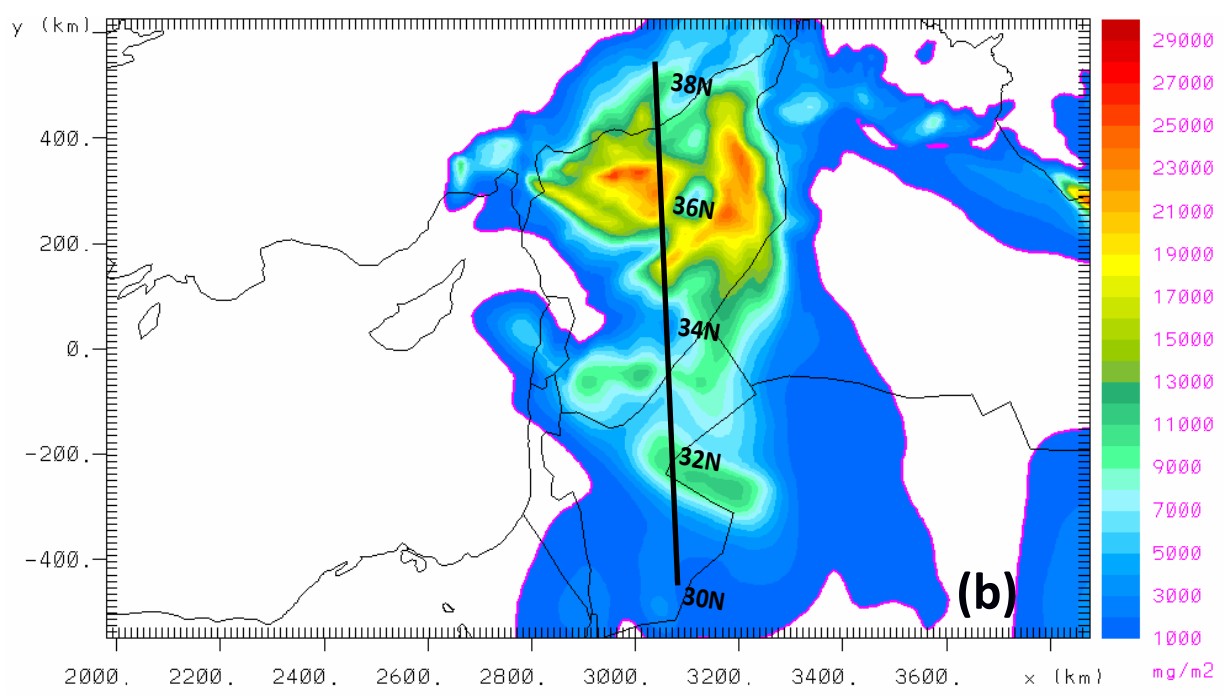


Figure 9. a) MSG-SEVIRI RGB map and CALIPSO overflight (green line) b) Model dustload (mg m$^{-2}$)

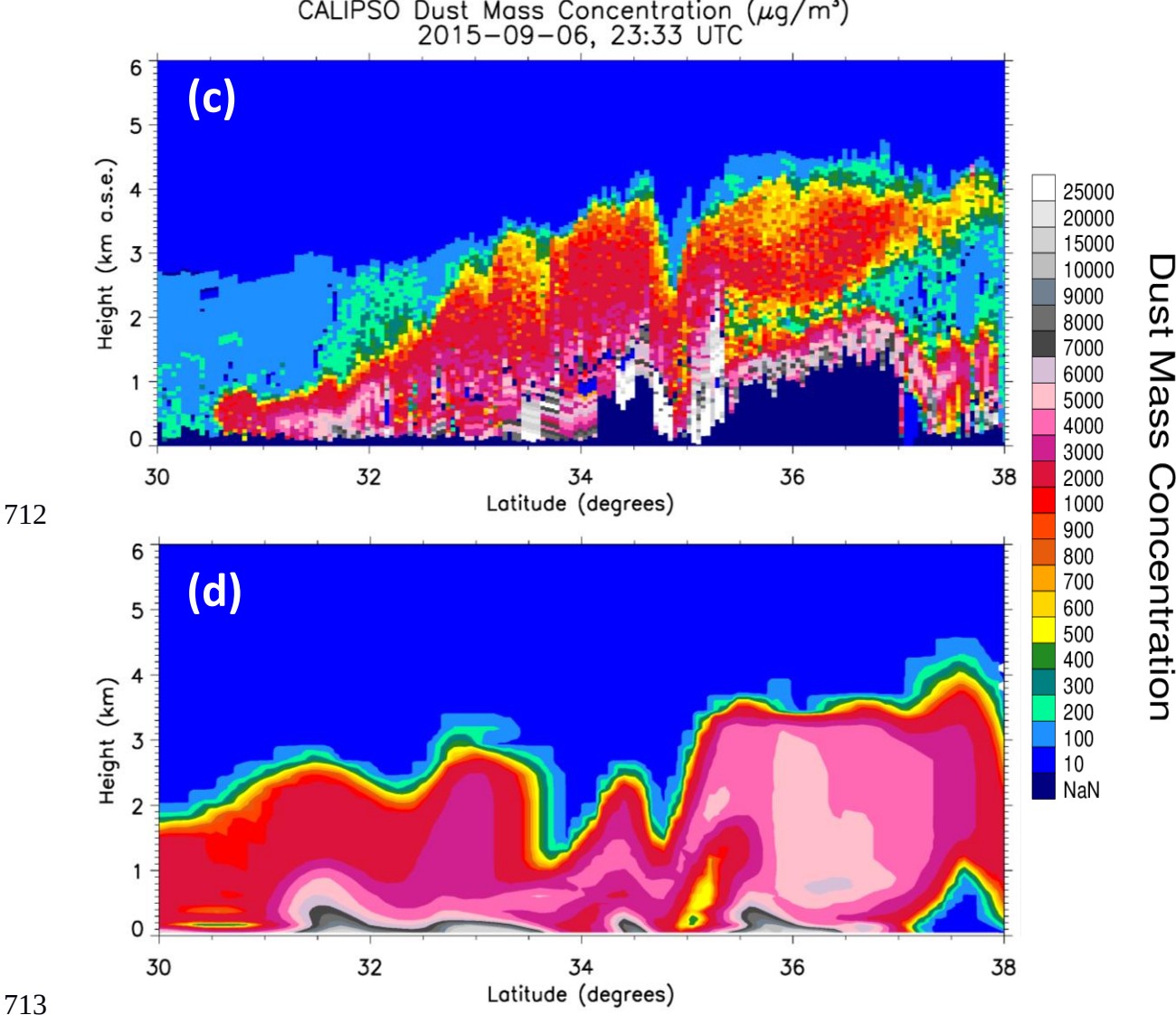



Figure 9. c) CALIPSO dust mass concentration (µg m$^{-3}$) and d) model dust mass concentration at 6

September 2015, 23:33 UTC. Due to the severity of the event CALIPSO signal is totally attenuated

bellow ~1km a.s.e. in the area between 35-37°N (dark blue color).



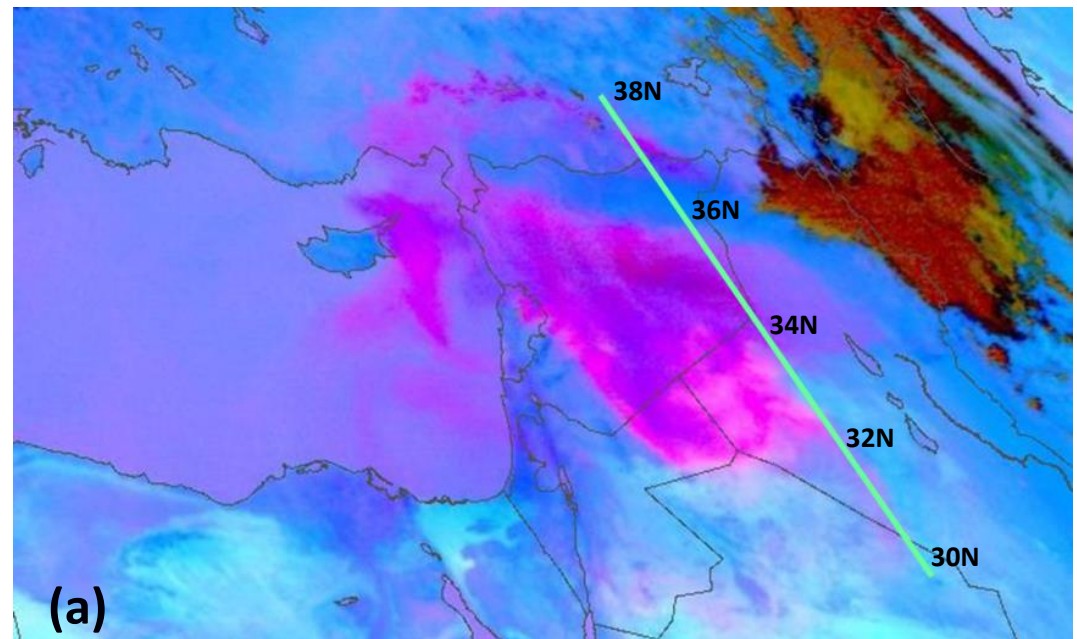


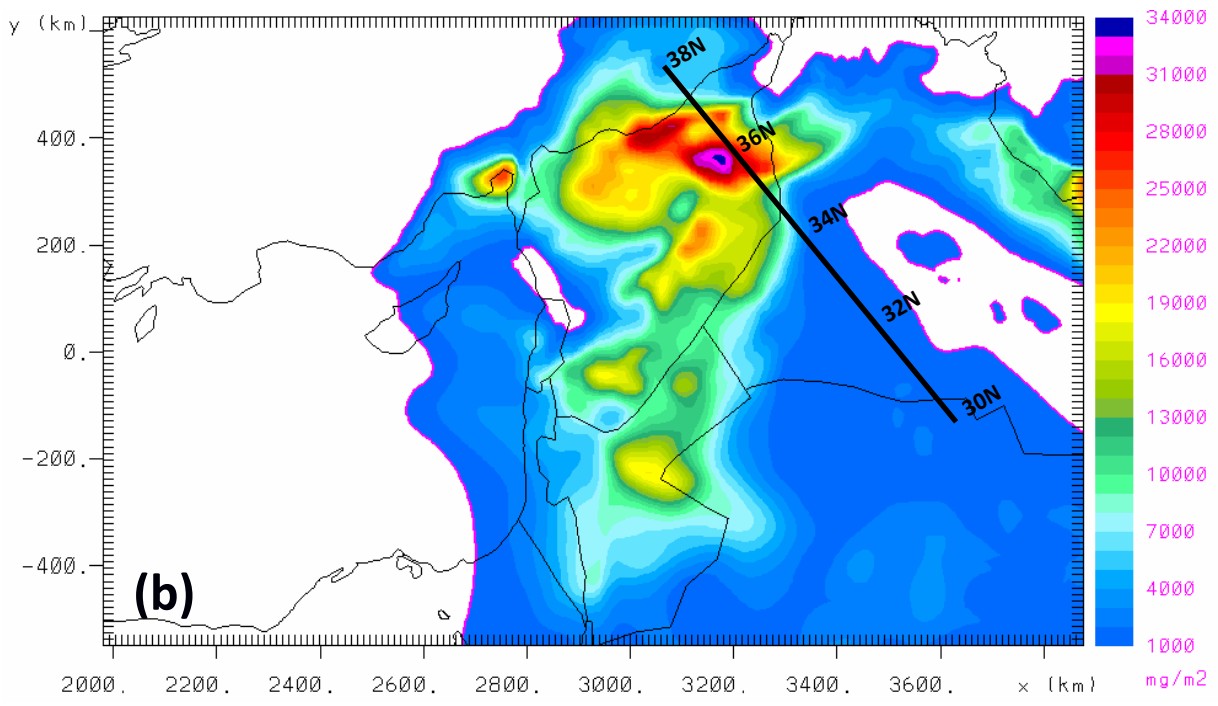

Figure 10. a) MSG-SEVIRI RGB map and CALIPSO overflight (green line), b) Model dustload (mg m$^{-2}$)

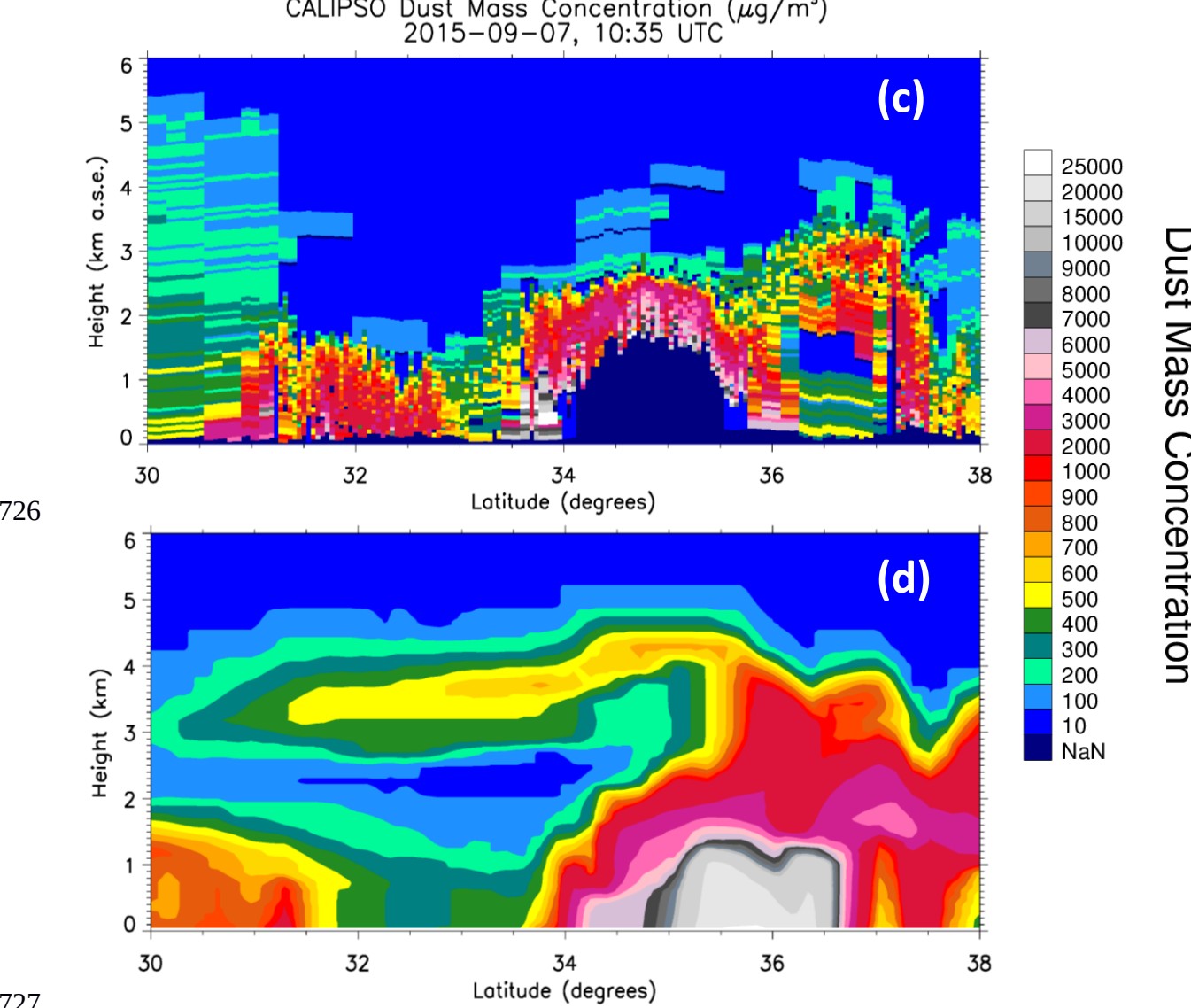



Figure 10. c) CALIPSO dust mass concentration (µg m⁻³) and d) model dust mass concentration at 7

September 2015, 10:35 UTC. Due to the severity of the event CALIPSO signal is totally attenuated

bellow ~1km a.s.e. in the area between 34-36°N (dark blue color).


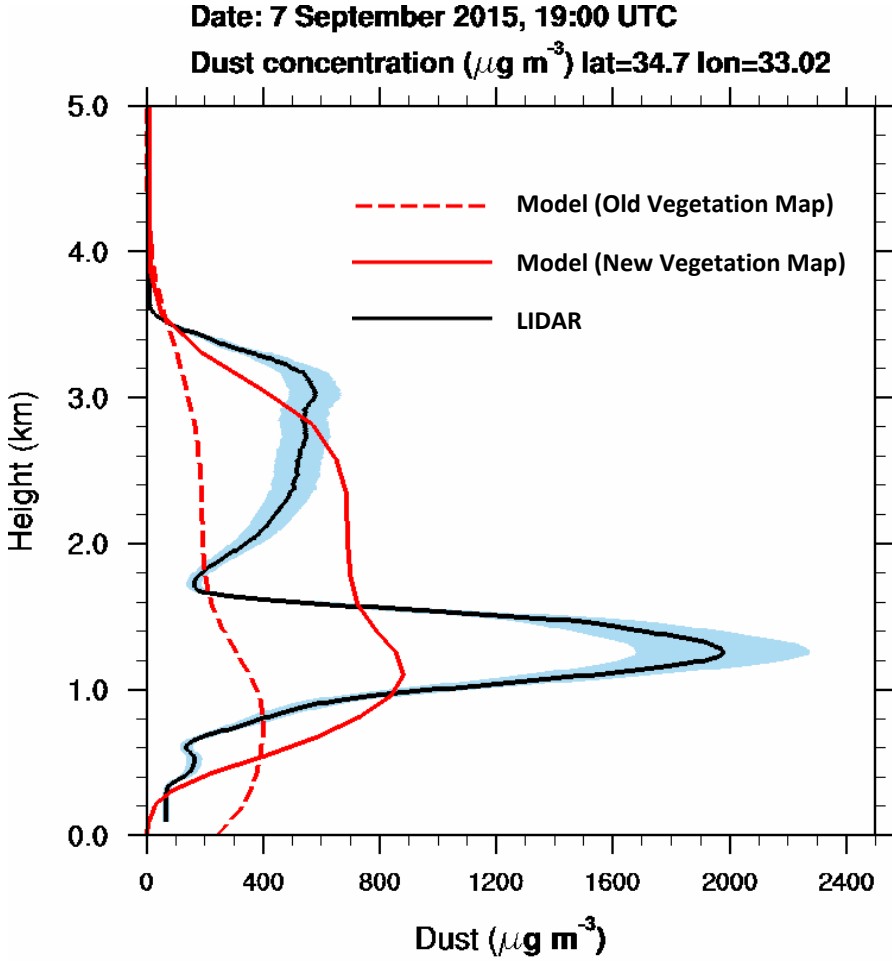

Figure 11. Vertical profile of dust concentration over Limassol on 7 September, 19:00 UTC. Blue
shadow indicates a 20% uncertainty of the lidar measurements.


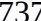




Figure12. Model 550 nm AOT over Cyprus 00:00 – 15:00 UTC, 8 September 2015, zoom from the second (4×4 km) model domain. The dashed black line shows the location of the cross-sections in Figure 13.


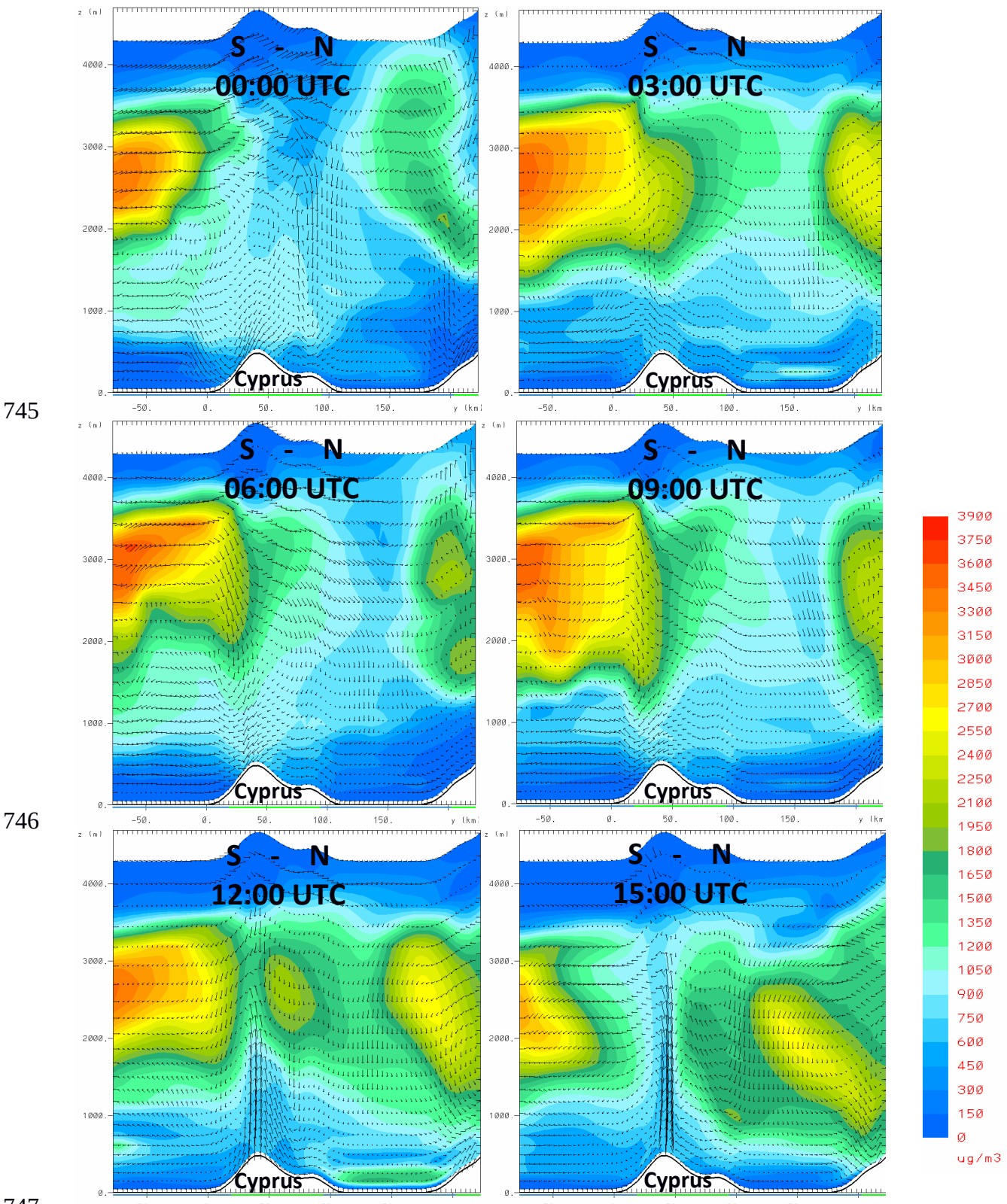




Figure 13. Vertical cross-section (South-North) of modeled dust concentration over Cyprus 00:00 –

15:00 UTC, 8 September 2015. The location of the cross-section is shown in Figure 12.
