# Peer review of "Remote sensing and modeling analysis of the extreme dust storm hitting Middle East and Eastern Mediterranean in September 2015"

_Atmospheric Chemistry and Physics, 2016_

## Referee Comment (RC1) · Anonymous Referee #3 · 20 Dec 2016

Review of "**Remote sensing and modelling analysis of the extreme dust storm hitting Middle East and Eastern Mediterranean in September 2015**"

Specific recommendations have been made by striking through text to be removed and additions are in red.

**Scientific significance** = 2 (1 if more were made of the importance of land surface changes with time and the quantitative impact of such changes on atmospheric dust loads.)

**Scientific quality** = 2

**Presentation quality** = 3 (Mainly based on difficulty of interpretation of figures)

Does the paper address relevant scientific questions within the scope of ACP? Yes

Does the paper present novel concepts, ideas, tools or data? Yes

Are substantial conclusions reached? Yes

Are the scientific methods and assumptions valid and clearly outlined? Mostly yes.

Are the results sufficient to support the interpretations and conclusions? Yes

Is the description of experiments and calculations sufficiently complete and precise to allow their reproduction by fellow scientists (traceability of results)? Yes

Do the authors give proper credit to related work and clearly indicate their own new/original contribution? Yes

Does the title clearly reflect the contents of the paper? Yes

Does the abstract provide a concise and complete summary? Yes

Is the overall presentation well structured and clear? Yes (some changes to figures required)

Is the language fluent and precise? Mostly yes

Are mathematical formulae, symbols, abbreviations, and units correctly defined and used? Not all, see below.

Should any parts of the paper (text, formulae, figures, tables) be clarified, reduced, combined, or eliminated? Yes, see below.

Are the number and quality of references appropriate? Yes

Is the amount and quality of supplementary material appropriate? N/A

**General comments**

This work is an interesting analysis of a dust storm that was generated over Iraq, Syria and Turkey and then produced a dust plume over the Eastern Mediterranean. The authors approach seems robust and the analysis of the meteorology leading to this dust event is in line with both literature and what I would expect from briefly looking into this particular case. It is generally well written and the figures show what the authors describe.

I think more could be made of the conclusion that changes to land surface over relatively short timeframes can be very important for specific dust events (and presumably the overall dust load). Especially with respect to the known interannual variability of particular dust sources such as ephemeral lakes and fluvial deposits from flooding.

**Specific comments**

45-51 Description of cold pool/haboob production could be made clearer. e.g.

" Haboobs are formed by the evaporation (and melting) of hydrometeors as they fall through warm, unsaturated air below the cloud base of convective clouds. The energy required for these phase changes (latent heat) generates cooled downdrafts. When the downdrafts hit the surface they spread out due to their enhanced density compared with the ambient air.  The convective outflow boundaries are turbulent and gusty and when they travel over bare soil and desert areas  sediment can be lifted, creating a propagating dust wall."

66 Not all cited works relate to the Atlas mountains remove "as an aftermath of Atlas Mountains convective storms."

125-131 I think section 2.1.3 would benefit from a bit more detail. In particular the production and limitations of the SEVIRI RGB dust images. I would like to direct the authors to the work of Banks & Brindley (2013) (http://dx.doi.org/10.1016/j.rse.2012.07.017) as an evaluation and description of the RGB SEVIRI dust images. Particularly the sensitivity to atmospheric moisture and dust height which is relevant to the interpretation of the SEVIRI dust images when dust is raised at different times and by different processes (recently lifted haboob dust less clearly distinguishable due to high atmospheric water vapour and high stability keeping dust close to the surface). This would be useful to add to the discussion in lines 225-227.

143 I am not familiar with the RAMS model but I suspect the levels are terrain following close to the surface but relax to be smooth and parallel in the upper levels. A little more detail would be useful here.

157 and figure 1. I think it would be useful to mark on the locations of the radiosonde launch stations on to Figure 1b. Also the frequency of the launches.

160-162 Did you use any different data for initialisation as part of your sensitivity studies? Roberts et al.(2014) (doi:10.1002/2013JD020667) and Schepanski et al. (2015) doi:10.1002/qj.2453 both show that over West/North Africa the data used for initialisation has a much larger impact on the resultant simulations than other factors such as model resolution, boundary layer set-up or microphysical schemes.

196-197 You mention the combination of cold air aloft and low level warming leading to a thermal low. I think it would be better to discuss and even show the 1000-700 hPa thickness and either mean sea level pressure or 925 hpa geopotential height to identify the formation of a thermal low.

238-243 I don't agree with this conclusion borne from Figure 7b that the temperature depression between the rain drops and the ambient air is the "crucial parameter". This is only a single factor that is likely to lead to high evaporation rates and therefore a strong cold pool. Arguably more important is the sub-saturation and depth of the below-cloud air. If below-cloud air is close to saturation and is shallow then regardless of the raindrop-ambient air temperature depression a strong cold pool will not be formed e.g. the patches over Turkey and Syria with similar values but no cold pool. The quantity of water held as hydrometeors is also important. Please amend to make it clear that the situation is more complex.

251-252 "This latency between satellite and modeled haboob fronts is an indication that the convective downdrafts were in fact stronger." Or could this also be attributed to a triggering delay due to the imperfect modelling of the boundary layer or the stability/moisture profile making conditions for triggering less favourable than in reality. Regardless of cold pool strength these factors could produce later triggering of convection and a latency in the storm progression compared with satellite imagery.

316-321 It looks like these are thermally driven downslope/upslope winds caused by preferential cooling/heating of the land surface compared to the surrounding sea.

327-329 There are many factors that could (and likely are) responsible. I think you should include a few more of them here e.g. fall speeds, limitations due to dust emission size bins, transport effects due to imperfect modelling etc.

**Technical corrections**

I feel that that work would have benefitted from being proof read by someone who is a native English speaker. There are occasions when slight errors disrupt the flow of the text. I have highlighted errors that I have seen below.

For example.

42-43 "These systems are well known by local populations  in desert and arid areas worldwide due to their devastating impact  on human health"

64 "A variety of studies on haboobs  have been performed worldwide."

Make sure acronyms are always defined where they first appear in the text.

For example.

55 "SAMUM 1 & 2" is not defined. SAharan Mineral dUst experiMent.

56 "MODIS"

105 "CALIOP" CALIPSO is defined but not CALIOP

Use "led" not "lead" throughout e.g. 66 and 89.

Wherever UTC time is used it would be useful for interpretation to include local time (LT) in brackets afterwards.

Be consistent with use of AOD or AOT, they are interchangeable.

53 "Moreover, haboobs are usually generated over remote …"

58 "It is also worth  mentioning …"

71 "synergy" I don't really like this term here. I'm not sure the effect is greater than the sum of the individual parts. If you are talking about specific positive feedbacks then be specific. If not you can just remove "and synergy"

114 "km  and vertical resolution…"

144 "dust  emission scheme"

187-188 "the combination  of two distinct meteorological features :"

202 "extended bare soil areas in Syria (Figure 2)."

213 " Figure 4 shows the convergence…"

221 "(plume_2)  was detected…"

223 "the approach  of the…"

233-234 "circulation and  is shown in Figure 6b.  It is characterized…"

234 "Somali"

259 "Figures  9 and  10"

272 "below"

281 "observations again suffers  from total…"

296 " Lebanon"

367 "regarding the forecast skill of the atmospheric…"

368 "such extreme episodes are  seldom, they still…"

370 "atmosphere are  now often adequately…"

374 "systems for dust episodes in West Africa."

375 "the complexity of these events makes  forecasting them very …"

Figures

Figure 1a Change the scale used here. I don't understand why you would only use the lowest third of the values specified on a colourbar. Label countries (at least Syria, Turkey and Iraq) for ease of interpretation.

Figure1b Include location of radiosondes that are assimilated. Their influence is obviously limited to a certain distance and time from the launch so knowing their position and frequency is important.

Figures 3, 5, 8, 9 & 10. Where possible SEVIRI RGB images should be cropped closer to the model domain they are compared with, either this or show more of the region and draw the domain box on top of the satellite imagery (Figure 5 was especially difficult to interpret) as the different panels are zoomed in different amounts and the model domains are rotated compared with the satellite imagery.

Figure 4. I think that it would be better for interpretation if the style and parameters plotted were changed slightly. Currently topography over 900 m is shaded. In reality we don't need this detail. You also discuss convergence but do not calculate or show it and interpreting convergence from wind vectors is very difficult. I suggest that you keep the vectors and the red contours for cloud (maybe make the contour lines thicker), but change the topography contours to a single blue or green contour at 1000 m. Then use colour filled contours (or greyscale) to show convergence. This can be as simple as a centred finite difference approach to show where the important convergence zones are.

Figure 6b Mark on location of domain shown in 6a.

Figure 7 As discussed in **Specific comments** I do not agree that the rain droplet to air temperature difference is the crucial parameter in the formation of the cold pool shown. Change 7a to be of colour contours of boundary layer sub-saturation or dew-point depression and have line contours of the rain droplet –air temperature difference  overlaid on top. This would show where the sub-saturation was strongest as well as where the temperature difference is greatest and where there are hydrometeors present.

Figures 9 and 10 should have an additional panel added that shows the model dust load marked with the cross section (equivalent to 9a and 10a). This would help with interpretation, especially given the delay in triggering of convection discussed in the paper.

---

## Referee Comment (RC2) · Anonymous Referee #4 · 5 Jan 2017

[referee-annotated manuscript omitted]

---

## Short Comment (SC1) · 7 Feb 2017

Dear colleagues,

I would like to draw your attention to our recently submitted manuscript. We simulated and analysed the same September 2015 event at convection permitting resolution with ICON-ART. Our results show a substantially different sequence of event stages compared to your findings. We are able to identify three consecutive cold pool outflows from two meso-scale convective systems, a finding which is confirmed by satellite observations. Furthermore, we think that an aerosol optical depth greater than 25, as it is visible in parts of your model domain, is unrealistic in both its magnitude as well as spatial distribution. You can find our work and explanations at

http://www.atmos-chem-phys-discuss.net/acp-2017-11/

We hope this information will contribute to the further understanding of the event.

Best regards, Philipp Gasch

——————————————————————

---

## Author Comment (AC2) · 23 Feb 2017

**1 Remote sensing and modeling analysis of the extreme dust storm hitting**

**2 Middle East and Eastern Mediterranean in September 2015**

3

4

- Solomos Stavros1, Albert Ansmann2, Rodanthi-Elisavet Mamouri3, Ioannis Binietoglou1,5, Platon
   Patlakas4, Eleni Marinou1,6 and Vassilis Amiridis1
- 7

[revised manuscript text omitted]

- 624
- 625

---

## Author Comment (AC3) · 23 Feb 2017

Thanks for your comment. The following sentences have been added at the conclusions section in the revised manuscript: "The key for forecasting these events in atmospheric models is the use of cloud resolving grid space. Forthcoming studies using an extended high-resolution grid over the entire Middle-East (e.g. Gasch et al., 2017) could provide more detail on the specific atmospheric processes during this episode".

The highest AOD values in the simulations are at the range 10-15. Extreme values > 20 are found over a few model grid points over the sources and due to the accumulation and overlapping of multiple dust layers. Such values are probably unrealistic; however the specific episode has indeed extraordinary characteristics.

[Figure]

Please also note the supplement to this comment:
http://www.atmos-chem-phys-discuss.net/acp-2016-1006/acp-2016-1006-AC3-
supplement.pdf

―――――――――――――――――――

**Supplement:**

[revised manuscript text omitted]

---

## Author Response (AR1)

**Authors Response for the manuscript: "Remote sensing and modeling analysis of the extreme dust storm hitting Middle East and Eastern Mediterranean in September 2015" by Stavros Solomos et al.**

**Response to Reviewer #3**

**General comments**
**This work is an interesting analysis of a dust storm that was generated over Iraq,Syria and Turkey and then produced a dust plume over the Eastern Mediterranean. The authors approach seems robust and the analysis of the meteorology leading to this dust event is in line with both literature and what I would expect from briefly looking into this particular case. It is generally well written and the figures show what the authors describe. I think more could be made of the conclusion that changes to land surface over relatively short timeframes can be very important for specific dust events (and presumably the overall dust load). Especially with respect to the known interannual variability of particular dust sources such as ephemeral lakes and fluvial deposits from flooding.**

We thank the reviewer for the thorough revision and comments. We agree with the importance of time-variant preferential dust sources due to changes in soil properties. The corresponding section has been extended in the revised manuscript and a new plot (Figure 3) has been added to comment on the land-use changes at the area of interest during the previous years. Replies to the specific comments follow below.

**Specific comments**

**45-51 Description of cold pool/haboob production could be made clearer. e.g.**
**"The responsible mechanism for haboob formation is the generation of a cold pool of ambient air due to evaporative cooling. The rain and ice condensates evaporate (or melt) as they fall through the warmer and unsaturated air and the absorption of latent heat from the phase changes leads in a vigor cooling of the surrounding air.**

**Haboobs are formed by the evaporation (and melting) of hydrometeors as they fall through warm, unsaturated air below the cloud base of convective clouds. The energy required for these phase changes (latent heat) generates cooled downdrafts. When the downdrafts hit the surface they spread out due to their enhanced density compared with the ambient air. When these The convective outflow boundaries are turbulent and gusty and when they travel over bare soil and desert areas they result in the generation of sediment can be lifted, creating a propagating dust wall."**

Done

**66 Not all cited works relate to the Atlas mountains remove "as an aftermath of Atlas Mountains convective storms."**

Done

**125-131 I think section 2.1.3 would benefit from a bit more detail. In particular the production and limitations of the SEVIRI RGB dust images. I would like to direct the authors to the work of Banks & Brindley (2013) (http://dx.doi.org/10.1016/j.rse.2012.07.017) as an evaluation and description of the RGB SEVIRI dust images. Particularly the sensitivity to atmospheric moisture and dust height which is relevant to the interpretation of the SEVIRI dust images when dust is raised at different times and by different processes (recently lifted haboob dust less clearly distinguishable due to high atmospheric water vapour and high stability keeping dust close to the surface). This would be useful to add to the discussion in lines 225-227.**

We have updated the corresponding section in the revised manuscript with a more detailed description of the product, and an extended discussion of its possible limitations.

**143 I am not familiar with the RAMS model but I suspect the levels are terrain following close to the surface but relax to be smooth and parallel in the upper levels. A little more detail would be useful here.**

The vertical coordinate system in RAMS is terrain following sigma-z and the grid stagger is Arakawa C. The first model level is at 50 m above ground and the levels stretch up to about 18 km. We have updated the relevant text accordingly.

**157 and figure 1. I think it would be useful to mark on the locations of the radiosonde launch stations on to Figure 1b. Also the frequency of the launches.**

Done.

**160-162 Did you use any different data for initialisation as part of your sensitivity studies? Roberts et al.(2014) (doi:10.1002/2013JD020667) and Schepanski et al. (2015) doi:10.1002/qj.2453 both show that over West/North Africa the data used for initialisation has a much larger impact on the resultant simulations than other factors such as model resolution, boundary layer set-up or microphysical schemes.**

Yes, but we found that our simulations are more sensitive to the location and dimensions of the inner grids rather than to initial conditions. More specifically inclusion of the 2×2 domain was the key to obtain model results that are closer to the observations. It is true that the simulation of the specific event could be further improved, however for the purpose of our study we believe that with the combination of remote sensing and modeling data all the principal processes driving this episode are adequately explained.

**196-197 You mention the combination of cold air aloft and low level warming leading to a thermal low. I think it would be better to discuss and even show the 1000-700 hPa thickness and either mean sea level pressure or 925 hpa geopotential height to identify the formation of a thermal low.**

Figure 3 is replaced by Figures 4b,c in the revised manuscript. Figure 4b shows the 1000-700 hPa thickness and Figure4c the 925 geopotential height, wind vectors and dust AOD. The corresponding

section has been revised in the manuscript.

**238-243 I don't agree with this conclusion borne from Figure 7b that the temperature depression between the rain drops and the ambient air is the "crucial parameter". This is only a single factor that is likely to lead to high evaporation rates and therefore a strong cold pool. Arguably more important is the sub-saturation and depth of the below-cloud air. If below-cloud air is close to saturation and is shallow then regardless of the raindrop-ambient air temperature depression a strong cold pool will not be formed e.g. the patches over Turkey and Syria with similar values but no cold pool. The quantity of water held as hydrometeors is also important. Please amend to make it clear that the situation is more complex.**

This is a good point and we agree with the reviewer on the importance of condensate mixing ratio and unsaturated air below the cloud base. The corresponding section has been extended in the revised manuscript, Figure 7a has been revised with the addition of relative humidity and two additional cross-section plots (Figures 7c,d in revised manuscript) have been added to indicate the severity of the particular convective storm. The cross section over the storm reveals the separate updraft-downdraft regions and a rainfall curtain extending from 4-5 km down to the surface. The cloud top is at 12 Km and the generation of a haboob is evident below the non-precipitating parts of the cloud.

**251-252 "This latency between satellite and modeled haboob fronts is an indication that the convective downdrafts were in fact stronger." Or could this also be attributed to a triggering delay due to the imperfect modelling of the boundary layer or the stability/moisture profile making conditions for triggering less favourable than in reality. Regardless of cold pool strength these factors could produce later triggering of convection and a latency in the storm progression compared with satellite imagery.**

We have rephrased this sentence accordingly: "The latency between satellite and modeled haboob fronts is possibly attributed to a slower propagating modeled haboob or to a triggering delay of convection in the model due to the imperfect representation of boundary layer properties and atmospheric stability."

**316-321 It looks like these are thermally driven downslope/upslope winds caused by preferential cooling/heating of the land surface compared to the surrounding sea.**

We agree and an extra sentence has been added in the revised text: "Differential heating between the land and water bodies and between flat terrain and mountain slopes results in the development of local wind flows (downslope / upslope winds)."

**327-329 There are many factors that could (and likely are) responsible. I think you should include a few more of them here e.g. fall speeds, limitations due to dust emission size bins, transport effects due to imperfect modelling etc.**

This sentence is rephrased in the revised manuscript: "(e.g. more intense downward mixing, increased emissions from the sources, limitations due to emission size bins, inaccurate deposition rates etc.)."

**Technical corrections**
**I feel that that work would have benefitted from being proof read by someone who is a native English speaker. There are occasions when slight errors disrupt the flow of the text. I have highlighted errors that I have seen below.**

We appreciate the thorough language review and we have corrected the text accordingly.

**For example.**
**42-43 "These systems are well known by local populations at in desert and arid areas worldwide due to their devastating impact in on human health"**
Done.

**64 "A variety of studies on haboobs has have been performed worldwide."**
Done.

**Make sure acronyms are always defined where they first appear in the text.  For example.**
**55 "SAMUM 1 & 2" is not defined. SAharan Mineral dUst experiMent.**
Done.

**56 "MODIS"**
Done.

**105 "CALIOP" CALIPSO is defined but not CALIOP**
Done.

**Use "led" not "lead" throughout e.g. 66 and 89.**
Done.

**Wherever UTC time is used it would be useful for interpretation to include local time (LT) in brackets afterwards.**

The experimental domain is quite extended and includes several time zones. We prefer to use UTC throughout the text for consistency.

**Be consistent with use of AOD or AOT, they are interchangeable.**

Done (AOT is used throughout the revised text).

**53 "Moreover, haboobs are usually generated over remote ..."**
**58 "It is also worth to mentioning..."**
**71 "synergy" I don't really like this term here. I'm not sure the effect is greater than the sum of the individual parts. If you are talking about specific positive feedbacks**

**then be specific. If not you can just remove "and synergy"**
**114 "km analysis and vertical resolution..."**
**144 "dust production emission scheme"**
**187-188 "the combination between of two distinct meteorological features in the greater area:"**
**202 "extended bare soil areas in Syria (Figure 2)."**
**213 "As seen in Figure 4, shows the convergence..."**
**221 "(plume_2) also was detected..."**
**223 "the approach approaching of the..."**
**233-234 "circulation and as is shown in Figure 6b. it It is characterized..."**
**234 "Somalia"**
**259 "Figures 8 9 and 9 10"**
**272 "bellow"**
**281 "observations again suffers again from total..."**
**296 "Libanon Lebanon"**
**367 "regarding the forecast skills of the atmospheric..."**
**368 "such extreme episodes are very seldom, they still..."**
**370 "atmosphere are nowadays now often adequately..."**

All spelling and grammar corrections are applied to the revised manuscript.

**374 "systems for dust episodes in West Africa."**

We have rephrased the sentence : "Moreover, a recent study by Pope et al. (2016) at the area of Sahel/southern Sahara suggests that unresolved haboobs during the summer monsoon may be responsible for up to 30% of the total atmospheric dust and such considerations raise questions on the current status of early warning systems for dust episodes."

**375 "the complexity of these events makes their forecast forecasting them very ..."**

Done

**Figures**

**Figure 1a Change the scale used here. I don't understand why you would only use the lowest third of the values specified on a colourbar. Label countries (at least Syria, Turkey and Iraq) for ease of interpretation.**

Done (Figure 4a in revised manuscript).

**Figure1b Include location of radiosondes that are assimilated. Their influence is obviously limited to a certain distance and time from the launch so knowing their position and frequency is important.**

Done (Figure 1 in revised manuscript).

**Figures 3, 5, 8, 9 & 10. Where possible SEVIRI RGB images should be cropped closer to the model domain they are compared with, either this or show more of the region and draw the domain box on top of the satellite imagery (Figure 5 was especially difficult to interpret) as the different panels are zoomed in different amounts and the model domains are rotated compared with the satellite imagery.**

As the reviewer also states the different projection between satellite and model images makes their intercomparison somehow tricky. We revised the aforementioned figures including indication of the model domain over the corresponding satellite images whenever possible.

**Figure 4. I think that it would be better for interpretation if the style and parameters plotted were changed slightly. Currently topography over 900 m is shaded. In reality we don't need this detail. You also discuss convergence but do not calculate or show it and interpreting convergence from wind vectors is very difficult. I suggest that you keep the vectors and the red contours for cloud (maybe make the contour lines thicker), but change the topography contours to a single blue or green contour at 1000 m. Then use colour filled contours (or greyscale) to show convergence. This can be as simple as a centred finite difference approach to show where the important convergence zones are.**

We have revisited this plot based on the reviewer's suggestions. Indeed no convergence zones are found. Moreover, the near surface wind field does not contribute to the transport of dust which occurs at levels above 1 km along with the convective outflow from the mountains of Turkey. Figure 4 is removed from the revised manuscript.

**Figure 6b Mark on location of domain shown in 6a.**

Done

**Figure 7 As discussed in Specific comments I do not agree that the rain droplet to air temperature difference is the crucial parameter in the formation of the cold pool shown. Change 7a to be of colour contours of boundary layer sub-saturation or dew-point depression and have line contours of the rain droplet –air temperature difference overlaid on top. This would show where the sub-**

**saturation was strongest as well as where the temperature difference is greatest and where there are hydrometeors present.**

Figure 7a has been revised following the reviewer's recommendations and two cross-section plots (Figures 7c,d) have been added to illustrate the severity of the convective storm and the generation of a density current haboob.

**Figures 9 and 10 should have an additional panel added that shows the model dust load marked with the cross section (equivalent to 9a and 10a). This would help with interpretation, especially given the delay in triggering of convection discussed in the paper.**

Done

**Response to reviewer #4**

We thank the reviewer for the comments and suggestions. All corrections are taken into consideration in the revised manuscript.

**Response to short comment**

Thanks for your comment. The following sentences have been added at the conclusions section in the revised manuscript: "The key for forecasting these events in atmospheric models is the use of cloud resolving grid space. Forthcoming studies using an extended high-resolution grid over the entire Middle-East (e.g. Gasch et al., 2017) could provide more detail on the specific atmospheric processes during this episode".

The highest AOD values in the simulations are at the range 10-15. Extreme values > 20 are found over a few model grid points over the sources and due to the accumulation and overlapping of multiple dust layers. Such values are probably unrealistic; however the specific episode has indeed extraordinary characteristics.

**List of changes**

Line 21:  and,

Line 22: as well as

Line 24: of modeling and remote sensing data

Line 28: Northern

Line 29: westward moving haboobs that merge

Line 30: Northern

Line 38: in this……… the Eastern

Line 39: Northern

Line 44: in desert

Line 45: on visibility

Lines 46-56: Haboobs are formed by the evaporation (and melting) of hydrometeors as they fall through warm, unsaturated air below the cloud base of convective clouds. The energy required for these phase changes (latent heat) generates cooled downdrafts. When the downdrafts hit the surface they spread out due to their enhanced density compared with the ambient air. These convective outflow boundaries are turbulent and gusty and when they travel over bare soil and desert areas sediment can be lifted, creating a propagating dust wall.

Line 59: haboobs are

Line 60: generated…….. in-site

Line 61-62: (e.g. SAharan Mineral dUst experiment (SAMUM)

Line 64-65: (e.g. Moderate Resolution Imaging Spectroradiometer (MODIS),

Line 66: Thickness (AOT),

Line 70: AOT

Line 72: have been…… For example

Line 74: that lead in severe haboob formation in Sahara.

Line 79: of the

Line 89: presented

Line 91: Therefore

Line 92: over Cyprus

Line 99: led

Line 106-108: The EARLINET lidar network is widely used for aerosol characterization and particularly for dust characterization studies (Mona et al., 2012).

Lines 110-112: For this case study we use a lidar ratio of 40 sr that is typical for Middle East dust

(Mamouri et al., 2013). The overall uncertainty in the estimated dust mass concentrations is 20-30%.

Line 116: The Cloud-Aerosol Lidar with Orthogonal Polarization (CALIOP),

Line 123-124: The CALIPSO algorithms are described in detail by Winker et al. (2009).

Line 125: resolution of 5 km and vertical

Lines 135-137: For this case study we use a lidar ratio of 40 sr that is typical for Middle East dust

(Mamouri et al., 2013). The overall uncertainty in the estimated dust mass concentrations is 20-30%.

Line 140: a combination of three infrared

Line 141: channels of SEVIRI

Lines 142-160: The channel combination and visualization parameters (Table 1) were chosen to maximize the visual contrast between the hot desert surface and lofted dust particles (Lensky and Rosenfeld, 2008). During daytime, the hot desert sand, made up from large quartz particles, appears white/blue due to the large difference in emissivity of IR10.8 and IR8.7 channels (green), high temperature (blue), and quite large difference in IR12.0 and IR10.8 channels (red). In contrast, lofted dust plumes with fine quartz particles have similar values of emissivity at IR10.8 and IR8.7 and this makes dust appear pink or magenta. Deep cumulonimbus clouds are depicted with red colors, while thick water clouds appear yellow. The RGB dust product is a very useful tool to qualitatively monitor dust transport events, taking advantage of the high temporal resolution of SEVIRI observations. The dust RGB product is provided in hourly intervals by EUMETSAT (European Organization for the Exploitation of Meteorological Satellites) and is used in this work to monitor the evolution of the dust transport event. In some cases, however, the usefulness of the product can be limited and this should be considered in the following discussion. First, the visual contrast of dust and the underlying surface is diminished when the temperature difference of the two is low, e.g. during nighttime. Second, high levels of columnar water vapor or the presence of the temperature inversion can mask the presence of dust in the atmosphere (Brindley et al., 2012). Finally, the contrast of dust and the ground can be further diminished over some type of surfaces e.g. over rocky terrain, due to its high emissivity at the 8.7 μm channel (Banks and Brindley, 2013).

Line 175: 50 sigma-z terrain

Line 176: The first model level is at 50 m above ground and the levels stretch from

Line 177: The dust emission scheme

Line 191: , 00Z and 12Z), …………., 00Z and 12Z

Line 192: , 00Z and 12Z…………, 21Z……………, 00Z

Line 199-200: hampered by seasonal and interannual variability of dust sources  together with  recent

Lines 202-204:Firstly, this annual-mean dataset cannot accurately describe the land use and dust sources at the end of the dry season in the Middle East., Secondly, the

Line 206: led to further changes

Line 208: between August

Lines 211- 224: The impact of the ongoing conflict on land use and vegetation can be further highlighted in Figure 3, showing the time series of MODIS NDVI in the region around Hawija, Kirkuk province, Iraq (region B of Fig.2). Agriculture in Hawija is based on a combination of rain-fed and

irrigated fields, in accordance with a rainy and a dry season from November to May and from June to October respectively. The NDVI time series clearly captures this behavior, with a major annual NDVI pick during wet months and a smaller cycle during each summer, probably reflecting the growth of summer crops with the help of irrigation. This summer cycle is completely absent in 2015. Indeed, a recent survey of the Food and Agriculture Organization (FAO) of the United Nations, reveals that large parts of the irrigation system in Kirkuk and surrounding regions have been destroyed by military operations and a large number of pumps and generators required for irrigation have been stolen (Singh N. et al, 2016). This, together with the destruction of other agricultural equipment and infrastructures, has severely disrupted the summer agriculture activities of 2015, exactly before the dust storm studied here, leaving the fields to act as very efficient dust sources.

Line 236-237: combination of two

Line 245-246: Advection of warm air from the Red Sea is also evident at the lowest troposphere by the 1000-700 mb thickness in Figure 4a.

Line 248: Syria that is evident by the 925 mb geopotential height in Figure 4b.

Line 255-258: and result in the mobilization of dust in the area. Dust uptake is mostly evident at the outer parts of the cyclone where surface wind speed exceeds

Line 262: Figure 4b

Line 263: Figure 4c

Lines 264-266: The convective outflows from the Zagros Mountains in Turkey that are evident by the black dashed line and wind vectors at 925 mb in Figure 3b enhance the mobilization of dust at the northern parts of the heat low.

Line 270: Cyprus is evident

Line 273: Mamouri et al.(2016). The faster propagating haboob plume (plume_2)

Line 276: In the model, approach of the haboob

Line 279: over Northern Syria and Southern Turkey

Line 283: on 6 September

Line 286: circulation and is

Line 287: Figure 6b. It is characterized by strong SW winds blowing from the Somali

Line 290-291: Low-level advection of warm towards the storm area also evident in Figure 6c by the 1000-700 mb modeled thickness at 15:00 UTC.

Line 293: A number of atmospheric  parameters that determine the formation of the cold

Line 294: are shown in Figures7a-d.

Line 295: iso-temperature line of -20°C  between

Line 297-299: clearly defines the cold pool area. Sub-saturated air below the cloud base is also evident in Figure 7a since the relative humidity at the neighbor of the convective cloud is between 15-20 %. The combination of sub-saturated air and

Line 300-301: and in the formation of a cold pool at the area of Northern Iraq with speeds ranging from

Line 302-308: The convective cloud top reaches 12 km and the updrafts exceed 18 m s$^{-1}$ at 15:00 UTC (Figure 7c). The rainfall curtain (downdraft area in Figure 7c) extends up to 4-5 km and the severity of the storm leads in the formation of a haboob that is evident by the streamline structure and dust production below the non-precipitating parts of the cloud in Figure 7d. A Kelvin-Helmholtz billow at 2-3 km separates the density current head from the ambient flow similar to previous findings for convective haboobs (Solomos et al., 2012). Dust particles are distributed inside the system and dust concentrations exceed 2000 μg m$^{-3}$ below the cloud. As the cold

Line 313: surface modelled

Lines 315-318: The latency between satellite and modeled haboob fronts is possibly attributed to a slower propagating modeled haboob or to a triggering delay of convection in the model due to the imperfect representation of boundary layer properties and atmospheric stability.

Line 326: Figures 9 and 10. All heights in satellite and model profiles refer to heights above surface

Line 328-329: The modeled dustload is also shown in Figure 9b for comparison. Dust concentrations are estimated from CALIPSO lidar signal

Line 330: Figure 9c they

Line 332: model (Figure 9d).

Line 340: totally attenuated below ~1 km

Line 342: at this area in the top 500m of the propagated haboob (1-1.5 km),

Line 345: The modeled dustload is also shown in Figure 10b for comparison

Line 348: (Figure 10c)

Line 350: observation again suffers

Line 352: from CALIPSO at the edge

Line 358: west of the CALIPSO ground track

Line 363: the lidar on 7 September

Line 365: off the coast of Lebanon

Line 373: On 8

Line 378: on 8 September

Line 383: during 8 September

Line 385-387: Differential heating between the land and water bodies and between flat terrain and mountain slopes results in the development of local wind flows (downslope / upslope winds).

Line 398: mixing, increased

Line 399: sources, limitations due to emission size bins, inaccurate deposition rates etc.).

Line 400: Table 2.

Line 416: Middle East and the Eastern

Line 422: of strong thermal low and convective outflows

Line 424: of moist and unstable air masses from the Arabian Sea and the Red Sea

Line 439: forecast skill

Line 440: are seldom, they

Line 442: are now often adequately

Line 444: Pope et al. (2016) at the area of Sahel/southern Sahara suggests

Line 445: haboobs during the summer monsoon

Line 448: makes forecasting them

Lines 449-458: As shown at the present study, the complexity of these events makes forecasting them very challenging and it is possible that a certain model configuration could successfully reproduce a specific event but not all similar events. The key for forecasting these events in atmospheric models is the use of cloud resolving grid space. However, such high resolution grid-space can only be applied over limited areas due to restrictions in computational power. Forthcoming studies using an extended cloud-resolving grid over the entire Middle-East (e.g. Gasch et al., 2017) could provide more detail on the individual atmospheric processes during this episode.

Line 465: activation of correlated

Line 496: Banks, J. R. and Brindley, H. E.: Evaluation of MSG-SEVIRI mineral dust retrieval products over North Africa and the Middle East, Remote Sensing of Environment, 128, 58–73, doi:10.1016/j.rse.2012.07.017, 2013.

Line 508: Brindley, H., Knippertz, P., Ryder, C. and Ashpole, I.: A critical evaluation of the ability of the Spinning Enhanced Visible and Infrared Imager (SEVIRI) thermal infrared red-green-blue rendering to identify dust

events: Theoretical analysis, J. Geophys. Res., 117(D7), D07201, doi:10.1029/2011JD017326, 2012.

Line 528: Gasch, P., Rieger, D., Walter, C., Khain, P., Levi, Y., and Vogel, B.: An analysis of the September 2015 severe dust event in the Eastern Mediterranean, Atmos. Chem. Phys. Discuss., doi:10.5194/acp-2017-11, in review, 2017.

Line 571: Mona L., Z. Liu, D. Müller, A. Omar, A. Papayannis, G. Pappalardo, N. Sugimoto and M. Vaughan, "Lidar Measurements for Desert Dust Characterization: An Overview," Advances in Meteorology, vol. 2012, Article ID 356265, 36 pages, 2012. doi:10.1155/2012/356265

Line 606: Singh N., van Zoonen D., and Khogir M.: Iraq agriculture and livelihoods needs assessment in the newly liberated areas of Kirkuk, Ninewa and Salahadin, Food and Agricutlure Organization of the United Nations, 2016.

Lines 644-649 : Table 1 …… Table 2

Line 656: Revised Figure 1.

Line 659: and black stars the location of radiosondes.

Line 678: New Figure 3.

Line 687: Revised Figure 4.

Line 687: a) Model 1000-700 mb thickness (dam), 6 September 2015, 00:00 UTC. Model AOT

Line 694: Remove old Figure 4

Line 705: Revised Figure 5

Line 706-707: The white rectangular approximately indicates the location of the model domain shown in Figure 5a.

Line 718: Revised Figure 6

Line 719: Wind speed at 975 mb

Lines 720-721: The white rectangular indicates the location of the model domain shown in Figure 6a.

c) Model 1000-700 mb thickness (dam), 6 September 2015, 15:00 UTC

Line 730: Revised Figure 7.

Lines 730-737: Figure 7. a) Model relative humidity at the first model level (color scale) and -20°C iso-temperature line (red contours) of rain droplets air temperature difference. b) Model wind speed at 10m (ms$^{-1}$). The dashed line denotes the location of the cold pool and the solid black line the location of the storm cross-sections of Figures 7c,d c) Vertical cross section of total condensate mixing ratio (blue contours in g k g$^{-1}$) and vertical wind component (vectors and color scale in m s$^{-1}$). The dashed line separates updraft (positive w) from downdraft/precipitating regions (negative w). d) Vertical cross section of total condensate mixing ratio (blue contours in g k g$^{-1}$), dust concentration (μg m$^{-3}$) and flow streamlines, 6 September 2015, 15:00 UTC

Line 741: Revised Figure 8

Lines 743-744: front location and the dashed rectangular in Figure 8c approximately indicates the location of the model domains shown in Figures 8a,b.

Line 751: Revised Figure 9.
Line 751: Model dustload (mg m$^{-2}$)  c)
Line 752: (µg m$^{-3}$) and d) model dust
Line 764: b) Model dustload (mg m$^{-2}$) c)

**Revised (marked-up manuscript version) with track changes**

[revised manuscript text omitted]

**RAMS 1000-700 hPa Thickness (dam)   6 September 2015 0 UTC**

[Figure]

[Figure]

Figure 3̶4. a) Model 1000-700 mb thickness (dam), 6 September 2015, 00:00 UTC. Model A̶O̶D̶ AOT (color scale) , geopotential height at 850 mb (red contours from 1490 to 1505 m every 2.5 m) and wind vectors at 850mb  ; b) MSG SEVIRI dust RGB at  08:00 UTC, 6 September 2015

[Figure]

Figure 4. Model topography (color scale), wind vectors at the first model layer (50 m) and cloud fraction > 70% (red contour), zoom from the second model grid, 6 September 2015, 00:00, 06:00, 12:00 UTC.

[Figure]

Figure 5. a) Model  AOT at 550 nm (color scale) and cloud cover > 70% (red contour). b) MSG-SEVIRI dust RGB component, 7 September 2015, 00:00 UTC. The white rectangular approximately indicates the location of the model domain shown in Figure 5a.

[Figure]

[Figure]

[Figure]

Figure 6. a) Model equivalent potential temperature (K) and wind vectors at 50m above ground, 6 September 2015, 13:00 UTC. b) Wind speed at  975 mb from the NCEP final analysis (FNL) dataset, 6 September 2015, 06:00 UTC. The white rectangular indicates the location of the model domain shown in Figure 6a. c) Model 1000-700 mb thickness (dam), 6 September 2015, 15:00 UTC

[Figure]

[Figure]

[Figure]

[Figure]

Figure 7. a) Model relative humidity at the first model level (color scale) and -20°C iso-temperature line (red contours) of rain droplets —air temperature difference in K. b) Model wind speed at 10m (ms⁻¹), 6 September 2015, 15:00 UTC. The dashed line denotes the location of the cold pool and the solid black line the location of the storm cross-sections of Figures 7c,d. c) Vertical cross section of total condensate mixing ratio (blue contours in g k g⁻¹) and vertical wind component (vectors and color scale in m s⁻¹). The dashed line separates updraft (positive w) from downdraft/precipitating regions (negative w). d) Vertical cross section of total condensate mixing ratio (blue contours in g k g⁻¹), dust concentration (μg m⁻³) and flow streamlines, 6 September 2015, 15:00 UTC

[Figure]

Figure 8. a) Model wind speed greater that 6 ms$^{-1}$ at 10 m and b) Near surface model dust concentration (μg m$^{-3}$) from the inner grid (2×2km) c) MSG-SEVIRI RGB component, 6 September 2015, 20:00 UTC. The dashed lines indicate the haboob front location and the dashed rectangular in Figure 8c approximately indicates the location of the model domains shown in Figures 8a,b.

[Figure]

[Figure]

Figure 9. a) MSG-SEVIRI RGB map and CALIPSO overflight (green line), Model dustload (mg m⁻²) b c) CALIPSO dust mass concentration (μg m⁻³) and cd) model dust mass concentration at 6 September 2015, 23:33 UTC. Due to the severity of the event CALIPSO signal is totally attenuated bellow ~1km a.s.e. in the area between 35-37°N (dark blue color).

[Figure]

[Figure]

[Figure]

Figure 10. a) MSG-SEVIRI RGB map and CALIPSO overflight (green line), b) Model dustload (mg m$^{-2}$) c) CALIPSO dust mass concentration (µg m$^{-3}$) and ed) model dust mass concentration at 7 September 2015, 10:35 UTC. Due to the severity of the event CALIPSO signal is totally attenuated bellow ~1km a.s.e. in the area between 34-36°N (dark blue color).

[Figure]

Figure 11. Vertical profile of dust concentration on 7 September 19:00 UTC over Limassol. Blue shadow indicates a 20% uncertainty of the lidar measurements.

[Figure]

Figure12. Model 550 nm AOT over Cyprus 00:00 – 15:00 UTC, 8 September 2015, zoom from the second (4×4 km) model domain. The dashed black line shows the location of the cross-sections in Figure 13.

[Figure]

Figure 13. Vertical cross-section (South-North) of modeled dust concentration over Cyprus 00:00 –
15:00 UTC, 8 September 2015. The location of the cross-section is shown in Figure 12a.

Table 1. Range and gamma correction for the Red, Green, and Blue channels for construct the Dust RGB product.

| Color | SEVIRI Channels | Min (K) | Max (K) | Γ |
|---|---|---|---|---|
| Red | IR12.0 – IR10.8 | -4 | 2 | 1 |
| Green | IR10.8 – IR8.7 | 0 | 15 | 2.5 |
| Blue | IR10.8 | 261 | 289 | 1 |